



# How concave are river channels?

Simon M. Mudd[1], Fiona J. Clubb[1,2], Boris Gailleton[1], and Martin D. Hurst[3]

[1]School of GeoSciences, University of Edinburgh, Drummond Street, Edinburgh EH8 9XP, UK
[2]Institute of Earth and Environmental Science, University of Potsdam, 14476 Potsdam-Golm, Germany
[3]School of Geographical and Earth Sciences, University of Glasgow, University Avenue, Glasgow G12 8QQ, UK

*Correspondence to:* Simon M. Mudd (simon.m.mudd@ed.ac.uk)

**Abstract.** For over a century geomorphologists have attempted to unravel information about landscape evolution, and processes that drive it, using river profiles. Many studies have combined new topographic datasets with theoretical models of channel incision to infer erosion rates, identify rock types with different resistance to erosion, and detect potential regions of tectonic activity. The most common metric used to analyse river profile geometry is channel steepness, or $k_s$. However,
the calculation of channel steepness requires the normalisation of channel gradient by drainage area. This relationship between channel gradient and drainage area is referred to as channel concavity, and despite being crucial in determining channel steepness, is challenging to constrain. In this contribution we compare both slope–area methods for calculating concavity and methods based on integrating drainage area along the length of the channel, using so-called "chi" ($\chi$) analysis. We present a new $\chi$-based method which directly compares $\chi$ values of tributary nodes to those on the main stem: this method allows us to
constrain channel concavity in transient landscapes without assuming a linear relationship between $\chi$ and elevation. Patterns of channel concavity have been linked to the ratio of the area and slope exponents of the stream power incision model ($m/n$): we therefore construct simple numerical models obeying detachment-limited stream power and test the different methods against simulations with imposed $m$ and $n$. We find that $\chi$-based methods are better than slope–area methods at reproducing imposed $m/n$ ratios when our numerical landscapes are subject to either transient uplift or spatially varying uplift and fluvial erodibil-
ity. We also test our methods on several real landscapes, including sites with both lithological and structural heterogeneity, to provide examples of the methods' performance and limitations. These methods are made available in a new software package so that other workers can explore how concavity varies across diverse landscapes, with the aim to improve our understanding of the physics behind bedrock channel incision.

## 1 Introduction

Geomorphologists have been interested in understanding controls on the steepness of river channels for centuries. In his seminal *Report on the Henry Mountains*, Gilbert (1877) remarked that: "We have already seen that erosion is favored by declivity. Where the declivity is great the agents of erosion are powerful; where it is small they are weak; where there is no declivity they are powerless." Following Gilbert's pioneering observations of landscape form, many authors have attempted to quantify how topographic gradients (or declivities) relate to erosion rates. Landscape erosion rates are thought to respond to tectonic uplift
(Hack, 1960). Therefore, extracting erosion rate proxies from topographic data provides novel opportunities for identifying



regions of tectonic activity (e.g., Seeber and Gornitz, 1983; Snyder et al., 2000; Lague and Davy, 2003; Wobus et al., 2006a; Cyr et al., 2010), and may even be able to highlight potentially active faults (e.g., Kirby and Whipple, 2012). Analysing channel networks is particularly important for detecting the signature of external forcings from the shape of the topography, as fluvial networks set the boundary conditions for their adjacent hillslopes, therefore acting as the mechanism by which climatic and

tectonic signals are transmitted to the rest of the landscape (e.g., Burbank et al., 1996; Whipple and Tucker, 1999; Whipple, 2004; Hurst et al., 2013).

Channels do not yield such information easily, however. Any observer of rivers or mountains will note that headwater channels tend to be steeper than channels downstream. Declining gradients along the length of the channel leads to river longitudinal profiles that tend to be concave up. Therefore, the gradient of a channel cannot be related to erosion rates in

isolation: some normalising procedure must be performed. Over a century ago Shaler (1899) postulated that as channels gain drainage area their slopes would decline, hindering their ability to erode. Beginning in the middle of the twentieth century authors such as Hack (1957), Morisawa (1962), and Flint (1974) pushed this idea further. Based on the hypothesis that a channel's capacity to carry water is likely to influence its erosive potential, these authors began to quantify the relationship between slope and drainage area, which is often used as a proxy for discharge.

Flint (1974) found that channel gradient appeared to systematically decline downstream in a trend that could be described by a power law:

$$S = k_s A^{-\theta}, \tag{1}$$

where $\theta$ is referred to as the concavity since it describes how concave a profile is: the higher the value, the more rapidly a channel's gradient decreases downstream. The term $k_s$ is called the steepness index, as it sets the overall gradient of the

channel. If we take the logarithm of both sides of equation (1), we find a line in $\log[S]$–$\log[A]$ space with a slope of $\theta$ and an intercept (the value of $\log[S]$ where $\log[A] = 0$) of $k_s$. A similar power-law relationship between slope and drainage area has been observed in channel profiles across the globe, and has been used by many authors to examine fluvial response to climate, lithology, and tectonics (e.g., Flint, 1974; Tarboton et al., 1989; Snyder et al., 2000; Kirby and Whipple, 2001; Lague and Davy, 2003; Wobus et al., 2006a).

## 1.1  Topography meets theory

In order to understand channel response to external forcings, the basic topographic relationship in equation (1) is often related to hypotheses predicting the relationship between channel incision and landscape properties such as drainage area and topographic gradient. Although topographic analysis does not depend on these hypotheses, interpretation of the results is often viewed through such a lens. There are a wide range of theories about functional relationships between channel properties and erosion





rates, but many can be represented by the general form of the so-called stream power incision model, first proposed by Howard and Kerby (1983):

$$E = K A^m S^n, \tag{2}$$

where $E$ is the long-term fluvial incision rate, $A$ is the upstream drainage area, $S$ is the channel gradient, $K$ is the erodibility coefficient, which is a measure of the efficiency of the incision process, and $m$ and $n$ are constant exponents. A number of variations of this equation are possible: some authors have proposed, for example, modifications that involve erosion thresholds (e.g., Tucker and Bras, 2000) or modulation by sediment fluxes (e.g., Sklar and Dietrich, 1998). However, Gasparini and Brandon (2011) showed that many of the modified versions of equation (2) could be captured simply by modifying the exponents $m$ and $n$. We can relate the stream power incision model to equation (1) by rearranging equation (2) for channel slope:

$$S = \left( \frac{E}{K} \right)^{-1/n} A^{-m/n}. \tag{3}$$

Comparing equations (1) and (3) reveals that the ratio between area and slope exponents in the stream power incision model, $m/n$, is therefore equivalent to the concavity, $\theta$, from equation (1). The channel steepness index, $k_s$, is related to erosion rate by:

$$k_s = \left( \frac{E}{K} \right)^{-1/n}. \tag{4}$$

Whipple and Tucker (1999) demonstrated that at steady state, defined here as where rock uplift rate, $U$, is equal to the erosion rate, $E$, both alluvial channels and bedrock channels should both exhibit the power law scaling of equation (3). In addition, Whipple and Tucker (1999) suggested that $m/n$ should fall in the range $0.35 \leq m/n \leq 0.6$ if bedrock incision is driven by shear stress. In channels that can be described by equation (2), the scaling between slope and area should also hold even if the landscape is transient: Royden and Perron (2013) demonstrated that changes in uplift rates can be transmitted upstream through channel networks as discrete "patches" where the local $k_s$ reflects local erosion rate.

The predicted relationship between the channel steepness index and uplift has been exploited by a number of studies to identify areas of tectonic activity (e.g., Kirby et al., 2003; Wobus et al., 2006a; Kirby and Whipple, 2012). Furthermore, many workers have used the framework of the stream power incision model to extract uplift histories (Pritchard et al., 2009; Roberts and White, 2010; Fox et al., 2014; Goren et al., 2014). However, the ability of these studies to extract information from channel profiles is dependent on the concavity index and the slope exponent, $n$, which are key unknowns within these theoretical models of fluvial incision. The concavity index is frequently assumed to be equal to 0.5, with $n$ assumed to be unity, despite recent compilations of data from multiple landscapes showing that this may not be the case (e.g., Lague, 2014; Harel et al., 2016; Clubb et al., 2016), and numerical modelling studies showing that $m/n$=0.5 leads to unrealistic relief structures (Kwang and Parker, 2017).





In this study we revisit commonly used methods for estimating the $m/n$ ratio using both slope–area analysis and methods that use channel elevations rather than channel gradients, often referred to as $\chi$ analysis, first introduced by Royden et al. (2000). Our objective is to determine the strengths and weaknesses of established methods alongside several new methods developed for this study, and to quantify the uncertainties in $m/n$ estimates. We present these methods in an open-source software package

that can be used to constrain channel concavity across multiple landscapes. This information may give insight into the physical processes responsible for channel incision into bedrock, which are as yet poorly understood.

## 2   Methods of constraining the $m/n$ ratio

### 2.1   Slope area analysis

The interpretation of erosion and uplift rates from river profiles is often performed by examining plots of channel slope against

drainage area (e.g., Snyder et al., 2000; Kirby and Whipple, 2001; Wobus et al., 2006a; DiBiase et al., 2010; Vanacker et al., 2015). In order to link slope and drainage area to erosion rate, we can take the logarithm of both sides of equation (3):

$$\log[S] = -m/n\log[A] + \log\left[\frac{E}{K}^{-1/n}\right],\qquad(5)$$

If we plot an idealised channel profile in $\log[S]$–$\log[A]$ space and fit a linear regression through the data, the gradient of the resulting line reflects the $-m/n$ ratio, and the intercept (where $\log[A] = 0$) reflects the erosion rate (or as shown by Royden

and Perron (2013), the local uplift rate if $n=1$). However, the gradient and the intercept from this regression will be correlated: therefore, to calculate the intercept and infer uplift rates, we assume an $m/n$ ratio that is constant throughout the profile and between different catchments. The intercept determined from this assumed $m/n$ ratio is often referred to as the normalised steepness index, $k_{sn}$, where the normalisation refers to fixing a value of $m/n$ (Wobus et al., 2006a):

$$k_{sn,i} = A_i^{m/n} S_i,\qquad(6)$$

where $i$ refers to individual locations in a channel network, and in equation (5) the same $m/n$ value is applied to every point in the channel network so that relative uplift or erosion rates can be inferred. As of the writing of this manuscript, dozens of papers have used slope-area analysis to infer uplift or erosion rates (e.g., Snyder et al., 2000; Kirby and Whipple, 2001; Kobor and Roering, 2004; Wobus et al., 2006a; Harkins et al., 2007; Cyr et al., 2010; DiBiase et al., 2010; Kirby and Whipple, 2012; Vanacker et al., 2015). Frequently this reference $m/n$ value is called $\theta_{ref}$, alluding to the fact that calculating concavity

from $\log[S]$–$\log[A]$ data requires no assumptions whatsoever about the underlying form of the equations describing channel incision: it is a purely geometric description of the channel profile. This is one advantage of $\log[S]$–$\log[A]$ methods over integral methods, described in the next section. To keep consistency between our descriptions of $\log[S]$–$\log[A]$ analysis and integral analysis we henceforth refer to reference $m/n$ ratios rather than interchanging $m/n$ and $\theta$.



The choice of the reference $m/n$ ratio is important in determining the relative $k_{sn}$ values amongst different sections in the channel network, which we illustrate in Figure 1. This figure depicts hypothetical slope–area data, which appear to lie along a linear trend in slope–area space. Choosing a reference $m/n$ based on a regression through these data will result in the entire channel network having similar values of $k_{sn}$. Based on the data in Figure 1, there is no evidence that the correct $m/n$ ratio is

anything other than the one represented by the linear fit through the data. However, these hypothetical data are in fact based on numerical simulations, presented in Section 3, in which we simulated a higher uplift rate in the core of the mountain range. The correct $m/n$ ratio is therefore lower than that indicated by the $\log[S]$–$\log[A]$ data, and instead the data show a strong spatial trend in channel steepness (interpretation 2 in Fig. 1). The simplest interpretation based on $\log[S]$–$\log[A]$ data alone would have been entirely incorrect. This situation is analogous to the one described by Kirby and Whipple (2001), where downstream

reductions in uplift rates in the Siwalik Hills of India and Nepal resulted in elevated apparent concavities. These examples highlight that selecting the correct $m/n$ ratio is crucial if we are to correctly interpret channel steepness data.

Furthermore, extracting the correct $m/n$ ratio from slope–area data on real landscapes is challenging: topographic data can be noisy, leading to a wide range of channel gradients for small changes in drainage area. The branching nature of river networks also results in large discontinuities in drainage areas where tributaries meet, resulting in significant data gaps in S-A

space (Figure 2). Wobus et al. (2006a) made recommendations for preprocessing of slope–area data that are still used in many studies: first, the DEM is smoothed, then topographic gradient is measured over either a fixed reach length or a fixed drop in elevation (Wobus et al. (2006a) recommends the latter), and then the data are averaged in logarithmically spaced bins. More recently, authors have proposed alternative channel smoothing strategies (e.g., Aiken and Brierley, 2013; Schwanghart and Scherler, 2017): all these proposed methods use some form of smoothing and averaging.

Here we forgo initial smoothing of the DEM and use a fixed elevation drop along a D8 drainage pathway implemented using the network extraction algorithm of Braun and Willett (2013). We calculate the best-fit concavity using two different methods: i) concavity extracted from all slope–area ($S$–$A$) data (i.e., no logarithmic bins); and ii) concavity of contiguous channel profile segments with consistent $S$–$A$ scaling within the log-binned $S$–$A$ data of the trunk stream, calculated using the statistical segmentation algorithm described in Mudd et al. (2014). We report the different extracted concavities and their

uncertainties in the results below.

## 2.2 Integral profile analysis

The noise inherent in $S$–$A$ analysis prompted Leigh Royden and colleagues to develop a method that compares the elevations of channel profiles, rather than slope (Royden et al., 2000). Like $S$–$A$ analysis, this method aims to normalise river profiles for their drainage area, but rather than comparing slope to area, their method integrates area along channel length (Royden et al.,





2000; Perron and Royden, 2013). The form of this integration is guided by equation (2). To illustrate the method, we integrate equation (3), assuming spatially constant incision equal to uplift (steady-state) and erodibility:

$$z(x) = z(x_b) + \left(\frac{E}{K}\right)^{\frac{1}{n}} \int\limits_{x_b}^{x} \frac{dx}{A(x)^{\frac{m}{n}}}, \tag{7}$$

where the integration is performed upstream from an arbitrary base level location ($x_b$) to a chosen point on the river channel,

$x$. The profile is then normalised to a reference drainage area ($A_0$) to ensure the integrand is dimensionless:

$$z(x) = z(x_b) + \left(\frac{E}{KA_0{}^m}\right)^{\frac{1}{n}} \chi, \tag{8}$$

where the longitudinal coordinate $\chi$ is equal to:

$$\chi = \int\limits_{x_b}^{x} \left(\frac{A_0}{A(x)}\right)^{m/n} dx. \tag{9}$$

The longitudinal coordinate $\chi$ has dimensions of length. The $\chi$ coordinate is simply a derived function of topography; it

can be calculated regardless of whether the landscape obeys equation (2). It should also be noted that although equation (7) is derived from a steady state model of channel incision, Royden and Perron (2013) showed that the linear relationship between $\chi$ and elevation should hold in linear segments such that the local slope in $\chi$-elevation space should reflect local erosion rates in transient landscapes. In addition, the slope in $\chi$–elevation space, which Mudd et al. (2014) called $M_\chi$, is the same as the normalised steepness index, $k_{sn}$, if $A_0$ is unity (cf. equation (6) and the second term to the right of the equality in equation (8)).

### 2.2.1   Extracting $m/n$ from $\chi$ profiles

In addition to providing a less noisy alternative to $S$–$A$ analysis, integral analysis also provides an independent test of the correct $m/n$ ratio. As demonstrated by Perron and Royden (2013), if the $m/n$ ratio is selected correctly, the main channel and tributaries should collapse onto a single profile. Perron and Royden (2013) suggested that the best fit $m/n$ ratio could be found by deriving values of $\chi$ for a series of $m/n$ values, performing a linear regression on each plot of $\chi$ against elevation, and

identifying the $m/n$ ratio at which the $R^2$ value of the regression is highest. However, this method is restricted to homogeneous, steady-state landscapes: if an idealised landscape is experiencing transient uplift it will be composed of segments of different gradients in $\chi$-elevation space (e.g., Royden and Perron, 2013). Mudd et al. (2014) therefore developed a statistical technique for fitting segments to the $\chi$ profiles, and then comparing the collinearity of these segments. However, this segmentation method is computationally expensive, and each segment is an approximation of the actual profile data. Hergarten et al. (2016) proposed

an alternative method wherein all pixels in a channel network are sorted by increasing elevation, the sum of the differences in adjacent $\chi$ values in this ranked list are computed, and from this metric a disorder function is calculated.





Here we present several new methods of identifying collinear tributaries in $\chi$-elevation space in order to constrain the best fit $m/n$ values from fluvial profiles. Rather than fitting segments to the profiles, which is computationally expensive, we directly compare all the elevation data of the tributaries in each drainage basin to the main stem. This is not completely straightforward, however: because the $\chi$ coordinate integrates area and channel distance it is very unlikely that a pixel on a tributary channel

shares a $\chi$ coordinate with any pixel on the main stem. Instead, for every tributary pixel we compare the tributary elevation with an elevation on the main stem at the same $\chi$ computed with a linear fit between the two pixels with the nearest $\chi$ coordinates (Figure 3). We then calculate a maximum likelihood estimator (MLE) for each tributary. The MLE is calculated with:

$$MLE = \prod_{i=1}^{N} \exp\left[-\frac{r_i^2}{2\sigma^2}\right],\tag{10}$$

where $N$ is the number of nodes in the tributary, $r_i$ is the calculated residual between the elevation of tributary node $i$ and

the linear regression of elevation on the main stem, and $\sigma$ is a scaling factor, which we can remove from the product term:

$$MLE = e^{-(0.5N/\sigma^2)} \prod_{i=1}^{N} \exp\left[r_i^2\right].\tag{11}$$

For a given drainage basin, we can multiply the MLE for each tributary to get the total MLE for the basin, and we can do this for a range of $m/n$ values to calculate the most likely $m/n$. As can be seen in equation (11) the value of the MLE will decrease as $N$ increases, and in large datasets this results in MLE values below the smallest number that can be computed, meaning

that in large datasets MLE values can often be reported as zero. To counter this effect we increase $\sigma$ until all tributaries have non-zero MLE values. As $\sigma$ is simply a scaling factor, this does not affect which $m/n$ value is calculated as the most likely value once all tributaries have non-zero MLEs (see supplementary information).

There are two disadvantages to using equation (10) on all points in the channel network. Firstly, because the MLE is calculated as a product of exponential functions, each data point will reduce the MLE and so tributaries will influence MLE in

proportion to their length. Secondly, because we use all data we cannot estimate uncertainty when computing the most likely $m/n$ value. Therefore, we apply a second method to the chi-elevation data that mitigates these two shortcomings, which we call a "Monte-Carlo points" method. It is a "points" method because the MLE is evaluated for a fixed number of discrete points on each tributary, and it is a Monte-Carlo method because we repeatedly sample points at random locations over many iterations, building up a population of MLE values for each $m/n$ ratio.

For each iteration of the Monte-Carlo points method, we create a template of points in $\chi$ space, measured from the confluence of each tributary from the trunk channel (Figure 4). We start by selecting a maximum value of $\chi$ upstream of the tributary junction, and then separate this space into $N_{MC}$ nodes. We create evenly spaced bins between the maximum value of $\chi$ in the template, and then in each iteration randomly select one point in each bin. Using this template on each tributary, we calculate the residuals between the tributary and the trunk channel using equation (10). If, for a given tributary, a point in the template is

located beyond the end of the tributary then the point is excluded from the calculation of MLE. Figure 4 provides a schematic visualisation of this method.



We repeat these calculations over many iterations and for each $m/n$ ratio we compute the median MLE, the minimum and maximum MLE, and the first and third quartile MLE. We approximate the uncertainty range by first taking the most likely $m/n$ ratio (having highest median MLE value amongst all $m/n$ ratios tested). We then find the span of $m/n$ ratios whose third quartile MLE values exceed the first quartile MLE value of the most likely $m/n$ ratio (Figure 4).

One complication of using collinearity to calculate the most likely $m/n$ value is that occasionally one may find a hanging tributary (e.g., Wobus et al., 2006b; Crosby et al., 2007), which could occur for a variety of reasons, such as the presence of geologic structures or lithologic variability. A hanging tributary can skew the overall MLE values in a basin, so in each basin we test the MLE and RMSE values in each tributary for outliers and iteratively remove these outlying tributaries, testing for the most likely $m/n$ value on each iteration. However, we find that eliminating outlying tributaries has a minimal effect on

the most likely $m/n$ value. The other primary complication is that one must assume an $m/n$ value prior to performing the chi transformation (equation 9) and thus slope–area analysis may be more suited to detecting changes in $m/n$ within basins (e.g., Wang et al., 2017b). We suggest here an alternative approach of calculating $m/n$ using $\chi$ methods in many small basins to look for any systematic changes. Before we can perform such analyses, however, we much constrain our confidence in estimates of the $m/n$ value.

## 15   3   Testing on numerical landscapes

In real landscapes, we can only approximate the $m/n$ ratio based on topography or by using time series information on the evolution of channel profiles, with data of the latter being vanishingly rare. Therefore, to test the relative efficacy of our methods for extracting the $m/n$ ratio we first run each method on a series of numerically simulated landscapes in which the $m/n$ ratio is prescribed. We employ a simple numerical model, following Mudd (2016), where channel incision occurs based

on equation (2). For computational efficiency, we do not include any other processes (e.g., hillslope diffusion) within our model. The elevation of the model surface therefore evolves over time according to:

$$\frac{\partial z}{\partial t} = U - KA^m S^n, \tag{12}$$

where $U$ is the uplift rate. Fluvial incision is solved using the algorithm of Braun and Willett (2013), where the drainage area is computed using the D8 flow direction algorithm to improve speed of computation and the topographic gradient is calculated

in the direction of steepest descent. In our model, we perform a direct numerical solution of equation (12) where $n = 1$ and use Newton-Raphson iteration where $n \neq 1$. These simulations are performed using the MuddPILE numerical model (Mudd et al., 2017), first used by Mudd (2016). We set the north and south boundaries of the model domain to fixed elevations, whereas the east and west boundaries are periodic. Our model domain is 30 km in the $X$ direction and 15 km in the $Y$ direction, with a grid resolution of 30 m. This allows us to test the methods of estimating $m/n$ on several drainage basins in each model domain,

and at a resolution comparable to that of globally-available digital elevation models (DEMs).



## 3.1 Transient landscapes

In order to test the methods' ability to identify the correct $m/n$ value, we ran a series of numerical experiments with varying $m/n$ ratios: $m/n = 0.5$, $m/n = 0.35$, and $m/n = 0.65$. For each ratio, we also performed simulations with varying values of $n$, as the $n$ exponent has been shown to impact the celerity with which transient knickpoints propagate through the channel

network (Royden and Perron, 2013). Crucially, Royden and Perron (2013) showed that when $n$ is not unity, upstream propagating knickpoints will erase information about past base level changes encoded in the channel profiles. This may cloud selection of the correct $m/n$ ratio, but Lague (2014) and Harel et al. (2016) have suggested many, if not most, natural landscapes have evidence for an $n$ exponent that is not unity. Therefore we ran simulations with $n = 1$, $n = 2$, $n = 1.5$, and $n = 0.66$ for each $m/n$ ratio, varying $m$ accordingly (see supplementary information for details of each model run).

We initialised the model runs using a low relief surface that is created using the diamond-square algorithm (Fournier et al., 1982). We found this approach resulted in drainage networks that contained more topological complexity than those initiated from simple sloping or parabolic surfaces. Our aim was to test the ability of each method to extract the correct $m/n$ ratio without assuming that the landscapes were in steady state: therefore each simulation was forced with varying uplift through time, to ensure that the channel networks were transient.

Each model was run with a baseline uplift rate of 0.5 mm yr$^{-1}$, which was increased by a factor of four for a period of 15,000 years, then decreased back to the baseline for another 15,000 years. For the runs with $n = 2$ the cycles were set to 10,000 years, which was necessary to preserve evidence of transience, as knickpoints propagate more rapidly through the channel network as $n$ increases. Relief is very sensitive to model parameters and we found in numerical experiments that basin geometry was sensitive to relief, mirroring the results of Perron et al. (2008). We wanted modelled landscapes to have comparable relief

and similar basin geometry across our simulations, to ensure similar landscape configurations for different values of $m$, $n$ and $m/n$. We therefore calculated the $\chi$ coordinate and solved equation (8) to find the $K$ value for each modelled landscape that produced a relief of 200 meters at the location with the greatest $\chi$ value given an uplift rate of 0.5 mm yr$^{-1}$.

We analysed these model runs using each of the methods of estimating the best fit $m/n$ outlined in Section 2. We extracted a channel network from each model domain using a contributing area threshold of $9 \times 10^5$ m$^2$. We performed a sensitivity analysis

of the methods to this contributing area threshold (see supplementary information), and found that the estimated best-fit $m/n$ ratios were insensitive to the value of the threshold.

Drainage basins were selected by setting a minimum and maximum basin area, $9 \times 10^6$ and $4.5 \times 10^7$ m$^2$ respectively; these values were chosen so extracted basins represented a good balance between the number of extracted basins and the number of tributaries in each basin. Nested basins were removed, as were basins that bordered the edge of the model domain. We exclude

basins on the domain boundaries as the calculation of the $\chi$ coordinate for the integral profile analysis is dependent on drainage area, which may not be realistic at the edge of the domain. Elimination of basins on the edge of the DEM is essential for real landscapes, as a basin beheaded by raster clipping will have incorrect $\chi$ values and we wanted to ensure both simulations and analyses on real basins used the same extraction algorithms. For each basin, we identified the best fit $m/n$ ratio predicted in four ways (as described in the methods section): i) by regression of all $\chi$-elevation data; ii) using $\chi$-elevation data processed





by our method of sampling points with the Monte Carlo method; iii) regressing the concavity though all slope–area data; and iv) regressions through slope–area data for individual segments of the main stem.

Figure 5 shows the spatial distribution of the predicted $m/n$ ratio for a series of basins from these cyclic model runs, where $m/n$ = 0.35, 0.5, and 0.65, and $n = 1$. We also plot the $m/n$ ratio predicted for each basin from all methods with varying values of $n$, an example of which is shown in Fig. 6. Our modelling results show that for each value of $m/n$ ratio tested, the method using all $\chi$ data identifies the correct ratio for every basin in the model domain. The Monte Carlo approach provides an estimate of the error on the best-fit $m/n$ ratio for each basin: Fig. 6 shows that there is no error on the predicted $m/n$ ratio, meaning that an identical $m/n$ ratio is predicted with each iteration of the Monte Carlo approach. The slope-area methods, in contrast, show more variation in the predicted $m/n$ ratio for each value of $m/n$ and $n$ tested (Figs. 5 and 6). Furthermore, the segmented slope–area data show a higher uncertainty in the predicted $m/n$ ratio compared to the other methods. The results of the model runs for all values of $m/n$ and $n$ are presented in the supplementary information.

### 3.2 Spatially heterogeneous landscapes

Alongside these temporally transient scenarios, we also wished to test the ability of each method to identify the correct $m/n$ ratio in spatially heterogeneous landscapes, simulating the majority of real sites where lithology, climate, or uplift are generally non-uniform. Therefore we performed additional runs where $m/n$ = 0.5, $n = 1$, but $U$ and $K$ varied in space. We generated the model domains using the same diamond-square initial condition as the spatially homogeneous runs. For the run with spatially varying $K$, we calculate the steady-state value of $K$ required to produce a surface with a relief of 400 m and an uplift rate of 1 mm yr$^{-1}$ using the same method as for the previous runs. From this baseline value of $K$, we calculated a maximum $K$ value which is five times that of the baseline. We then created ten "patches" within the initial model domain where $K$ was assigned randomly between the baseline and the maximum.

For the spatially varying uplift run, we varied uplift in the N-S direction by modelling it as a half sine wave:

$$U = U_A \sin((\pi y)/L) + U_{min}, \tag{13}$$

where $y$ is the northing coordinate and $L$ is the total length of the model domain in the $y$ direction, $U_A$ is an uplift amplitude, set to 0.2 mm/yr, and $U_{min}$ is a minimum uplift, expressed at the North and South boundaries, of 0.2 mm/yr. Both scenarios, with spatially varying erodibility and uplift, were run to approximately steady state: the maximum elevation change between 15,000 year printing intervals was less than a millimeter.

Inherent in equation (7) is the assumption that $U$ and $K$ do not vary in space: our spatially heterogeneous experiments therefore violate basic assumptions of the integral method. These conditions, however, are likely true in virtually all natural landscapes. Therefore, our aim here was to test if we could recover $m/n$ ratios from numerical landscapes that are more similar to real landscapes than those with spatially homogeneous $U$ and $K$.

Figure 7 shows the distribution of predicted $m/n$ ratios for the runs with spatially varying $K$ and $U$ from both the integral Monte Carlo approach and the slope–area method. In comparison to our model runs where $K$ and $U$ were uniform, each method



performs worse at identifying the correct $m/n$ ratio of 0.5. However, in both model runs the integral methods identified the correct ratio in a higher proportion of the drainage basins than the slope-area methods. Furthermore, the distribution of $m/n$ predicted by the integral methods reaches a peak at the correct $m/n$ ratios of 0.5, suggesting that even in spatially heterogeneous landscapes the methods can still be applied. Our run with the random distribution of erodibility patches shows that the correct

calculation of the $m/n$ ratio is highly dependent on the spatial continuity of $K$: in basins contained within a single patch (e.g., basins 4, 5, and 6), the integral profile method correctly identified the $m/n$ ratios. Figure 8 shows example $\chi$-elevation plots at varying $m/n$ ratios for basin 2, which encompasses several patches with varying $K$ values. Within this basin, tributaries that drain a patch with the same $K$ value are still collinear in $\chi$-elevation space. Based on these results, we suggest that, in real landscapes, monolithologic catchments should be analysed wherever possible in order to select an appropriate $m/n$ ratio.

## 10  4   Constraining $m/n$ in real landscapes

Our numerical modelling results suggest that the integral profile analysis is most successful in identifying the correct $m/n$ ratio out of the entire range of $m/n$ and $n$ values tested. However, these modelling scenarios cannot capture the range of complex tectonic, lithologic, and climatic influences present in nature. Therefore, we repeat our analyses on a range of different landscapes with varying climates, relief structures, and lithologies, to provide some examples of the variation of $m/n$ ratios

predicted using each method. For each field site, topographic data were obtained from OpenTopography, using the seamless DEM generated from NASA's Shuttle Radar Topography Mission (SRTM) at a grid resolution of 30 m. The supplemental materials contain metadata for each site so readers can extract the same topographic data used here.

### 4.1   An example of a relatively uniform landscape: Loess Plateau, China

In order to demonstrate the ability of the methods to extract the $m/n$ ratio in a relatively homogeneous landscape, we first

analyse the Loess Plateau in northern China. The channels of the Loess Plateau are incising into wind-blown sediments that drape an extensive area of over 400,000 km$^2$ (Zhang, 1980), and can exceed 300 m thickness (Fu et al., 2017). The plateau is underlain by the Ordos block, a succession of non-marine Mesozoic sediments which has undergone stable uplift since the Miocene (Yueqiao et al., 2003; Wang et al., 2017a). Although there have been both recent (Wang et al., 2016) and historic (Wang et al., 2006) changes in sediment discharge from the plateau, the friable substrate means that channel networks and

channel profiles might be expected to adjust quickly to perturbations in erosion rate. Indeed, Willett et al. (2014) suggested, based on differences in the $\chi$ coordinate across drainage divides, that the channel networks in large portions of the plateau are geomorphically stable. The stable tectonic setting and homogeneous, weak substrate of the Loess Plateau makes an ideal natural laboratory for testing our methods on relatively homogeneous channel profiles.

We ran each of the methods on an area of the Loess Plateau approximately 11,000 km$^2$ in size near Yan'an, in the Chinese

Shaanxi province (Figure 9a). We find relatively good agreement between both the chi and slope–area methods of estimating the most likely $m/n$ ratio. Figure 9b shows the probability distribution of $m/n$ ratios determined from the population of the most likely $m/n$ ratio from each basin (i.e., it does not include underlying uncertainty in each basin), but the peaks of these curves lie



at an $m/n$ ratio of approximately 0.4 using both the Monte Carlo points method and all slope–area data, and at approximately 0.5 using the all $\chi$ data method. This level of agreement gives us some confidence that channel steepness analyses using reference concavities between 0.4 and 0.5 should give an accurate representation of the relative steepness of the channels. We ran each of the methods on an area of the Loess Plateau approximately 11,000 km$^2$ in size near Yan'an, in the Chinese Shaanxi

province (Figure 9a). We find relatively good agreement between both the chi and slope–area methods of estimating the most likely $m/n$ ratio. Figure 9b shows the probability distribution of $m/n$ ratios determined from the population of the most likely $m/n$ ratio from each basin (i.e., it does not include underlying uncertainty in each basin), but the peaks of these curves lie at an $m/n$ ratio of approximately 0.4 using both the Monte Carlo points method and all slope–area data, and at approximately 0.5 using the all $\chi$ data method. This level of agreement gives the worker some confidence that channel steepness analyses using

reference concavities between 0.4 and 0.5 should give an accurate representation of the relative steepness of the channels.

     As well as determining the best-fit $m/n$ value for the landscape as a whole, we can also examine the channel networks in individual basins: Figure 9c shows the $\chi$-elevation profiles for an example basin. In this basin the tributaries are well aligned with the trunk channel at the most likely $m/n$ ratio of 0.4, both using all the chi data and with the Monte Carlo approach. In our explorations of different landscapes, the Loess Plateau is the landscape that most resembles the idealised landscapes that

we find in our model simulations. The Loess Plateau is notable for the homogeneity of its substrate over a large area; most locations on Earth are not as homogeneous.

## 4.2   An example of lithologic variability: Waldport Oregon, USA

Many studies analysing the steepness of channel profiles are focused in areas where external factors, such as lithology or tectonics, are not uniform. Here we select an example of a landscape with two dominant lithologic types in a location along the

Oregon Coast near the town of Waldport, Oregon (Fig. 10). The Oregon Coast Ranges are dominated by the Tyee Formation, made up primarily of turbidites deposited during the Eocene (Heller et al., 1987). In addition to these sedimentary units, our selected landscape also contains the Yachats Basalt, which erupted mostly as subaeral flows between 3 and 9 meters in thickness during the late Eocene (Davis et al., 1995). Erosion rates inferred from $^{10}$Be concentrations in stream sediments are between 0.11 to 0.14 mm per year (Heimsath et al., 2001; Bierman et al., 2001), similar to rock uplift rates of 0.05-0.35 mm

per year inferred from marine terraces (Kelsey et al., 1994). Short term erosion rates derived from stream sediments fall into the range of 0.07 to 0.18 mm per year (Wheatcroft and Sommerfield, 2005), leading a number of authors to suggest that the Coast Ranges are in topographic steady state, where uplift is balanced by erosion (e.g., Reneau and Dietrich, 1991). Thus our site contains a clear lithologic contrast but has been selected to minimise spatial variations in uplift or erosion rates.

     We find that whereas basins developed on basalt have a relatively uniform $m/n$ of approximately 0.7, the most likely $m/n$

ratios in the sandstone show considerably more scatter (Figure 10b), with a lower average $m/n$ ratio. We present these data as an example of spatially varying $m/n$ as as a function of lithology; future workers could explore if weaker rock leads to higher concavity values as suggested by Duvall et al. (2004). The $\chi$ profiles in basin 17 (Figure 10c) are notable because this basin features two bedrock types: basalt in the lower reaches and sandstone in the headwaters. If the $m/n$ ratio is too high, tributaries will fall below the trunk channel in chi–elevation space. In Figure 10c, the $m/n$ ratio is chosen to reflect the typical value of



the basalt basins, and tributary channels in the sandstone fall below the trunk channel, meaning that changes in $m/n$ ratios can be seen within basins. This means that workers must be cautious when using a reference concavity or $m/n$ ratio in determining channel steepness indices in basins with heterogeneous lithology.

### 4.3 An example of a tectonically active site: Gulf of Evia, Greece

The steepness of channel profiles and presence of steepened reaches (knickpoints) in tectonically active areas can reveal spatial patterns in the distribution of erosion and/or uplift (e.g., Densmore et al., 2007; DiBiase et al., 2010; Vanacker et al., 2015) and has the potential to allow identification of active faults (e.g., Kirby and Whipple, 2012). However, these systematic spatial patterns in channel steepness may challenge our ability to constrain $m/n$. Our third example is in a tectonically-active landscape where we have found spatial variations in the most likely $m/n$ ratio between catchments proximal to active normal faults. We

explore a series of basins draining across faults in the Sperchios Basin, Gulf of Evia, Greece (Figure 11), predominantly cut into clastic sediments (Eliet and Gawthorpe, 1995). Previous work (Whittaker and Walker, 2015) has demonstrated that catchment morphology reflects interaction with these faults. The rivers are typically characterised by convex longitudinal profiles that commonly have two knickpoints. The upper set of knickpoints are attributed to the initiation of faulting and the resulting growth of topography. The lower set of knickpoints are interpreted as the result of subsequent increase (3-5×) in throw rate

due to fault linkage (Whittaker and Walker, 2015). The elevations of each group of knickpoints both scale with footwall relief, suggesting that fault throw rates scale with fault segment length.

Steep, smaller catchments tend to drain across the footwalls of these faults, whilst larger catchments drain the landscape behind the faults, through the relay zones between fault segments. We derived the $m/n$ ratios for each catchment following each of the four methods (Figure 12). Given the presence of knickpoints along the river profiles, it is not appropriate to derive

$m/n$ ratios by linear regression of all log[$S$]–log[$A$] data. We find that the $m/n$ ratios derived from segmented slope-area analysis are highly variable between catchments (Figure 12, inset), with a tendency toward abnormally large values, generally exceeding the upper range of values typically predicted by incision models (Whipple and Tucker, 1999). Values of $m/n$ derived using the $\chi$ methods are predicted to be relatively low, typically 0.1-0.6 (Figure 12), and whilst the two $\chi$ methods do not agree perfectly, they do co-vary, and are for the most part within uncertainty of each other (with the exception of basins 1 and 20).

Lowest values of $m/n$ = 0.1 typically occur for the small, steep catchments draining across the footwalls of the fault segments (e.g., Basin 10, Figure 13), with higher $m/n$ values typical for catchments that do not cross faults, or those that cross relay zones (e.g., Basin 7, Figure 13). Plots of $\chi$-elevation such as in Figure 13 demonstrate that there can be considerable variability in the morphology of tributaries as they respond to adjustment in the trunk channel.

Our aim here is not to provide a comprehensive examination of the topography and tectonic evolution of the Sperchios

Basin (see Whittaker and Walker, 2015) but to demonstrate the impact of tectonic transience on our ability to quantify $m/n$. Low values of $m/n$ in steep small catchments draining across the faults may reflect the contribution of debris flow processes to valley erosion at smaller drainage areas, which tends to lead to lower apparent $m/n$ ratios in the topography (Stock and Dietrich, 2003). Additionally, these catchments may in effect behave as fluvial hanging valleys (Wobus et al., 2006b). Values of $m/n$ derived using the Monte Carlo points method are in all cases equal to or lower than values derived using all $\chi$ data.



This is noteworthy because of the difference in how tributaries are weighted between the two techniques. Using all $\chi$ data, longer tributaries have more influence on the calculation of the most likely $m/n$, whereas the Monte Carlo points methods weights each tributary equally (since the same number of points are sampled on each tributary). Thus, if the steepness of the channels at low drainage area is influenced by debris flow processes (Stock and Dietrich, 2003), we would expect this to be
more influential on the derived $m/n$ when using the Monte Carlo points method, resulting in lower $m/n$ values.

Finally, it is recognised that transient landscapes are likely settings for drainage network reorganisation (Willett et al., 2014). In the absence of lithologic variability, climate gradients and tectonic transience, gradients in $\chi$ in the channel network between adjacent drainage basins are predicted to indicate locations where drainage divides are migrating (toward the catchment with higher $\chi$) and drainage network reorganisation is ongoing (Willett et al., 2014). Rivers draining across normal fault systems
are often routed through the relay zones between fault tips, where uplift rates are lowest, capturing and rerouting much of the drainage area above the footwall (e.g., Paton, 1992). In the Sperchios Basin this has resulted in strong gradients in $\chi$ across topographic divides (Figure 14), particularly between the large catchments draining the landscape behind the footwall (which have likely been gaining drainage area), and the short, steep catchments draining across the footwall (which have likely been truncated). Where catchments are growing or shrinking, relationships between $\chi$ and elevation are expected to
deviate systematically from a steady-state straight profile, with aggressor catchments having steeper $\chi$ profiles (resulting in higher apparent $m/n$ derived from topography) and victims having gentler $\chi$ profiles (lower $m/n$). This is consistent with our observations of low $m/n$ ratios in short, steep catchments draining across the footwall that may have lost drainage area during fault growth.

Our analysis of the topography in the Sperchios Basin, whilst not exhaustive, highlights that river profiles alone and the
resulting $m/n$ (and/or $k_{sn}$) derived from topography are not alone sufficient to interpret the history of landscape evolution, bust must be considered alongside other observational data and in the context of a process-based understanding of landscape evolution and tectonics.

## 5   Conclusions

For over a century, geomorphologists have sought to link the steepness of bedrock channels to erosion rates, but any attempt
to do so requires some form of normalisation. This normalisation is required because in addition to topographic gradient, the relative efficacy of incision processes is thought to correlate with other landscape properties that are a function of drainage area, such as discharge or sediment flux. Theory developed over the last four decades suggest that the channel concavity may be used to normalise channel gradient, and over the last two decades many authors have compared the steepness of channels normalised to a reference concavity derived from slope–area data (e.g., Snyder et al., 2000; Kirby and Whipple, 2001). In recent years an
integral method of channel analysis has also been developed (e.g., Perron and Royden, 2013) that can complement slope–area analysis and via alignment of tributaries provide an independent test of the most likely $m/n$ ratio of the channel network, which is related via stream power theory to channel concavity.





In this contribution we have developed a suite of methods to quantify the most likely $m/n$ ratio using both slope–area analysis and the integral method. In addition to traditional S–A methods, we also present methods of analysing $\chi$-transformed channel networks that do not require the profiles to be linear from source to outlet, but constrain the $m/n$ ratio based on quantifying the residuals between every node on each tributary and the trunk channel. In a second method we quantify uncertainty on the

predicted value of $m/n$ using a subset of points on the tributary network that are randomly assigned within a Monte-Carlo sampling framework. We then test these methods against idealised, modelled landscapes that obey the stream power incision law but have been subject to transient uplift, as well as spatially varying uplift and erodibility.

We find that $\chi$-based methods are best able to reproduce the $m/n$ ratios imposed on the model runs. The most likely $m/n$ ratios determined from $\chi$-based methods on transient landscapes have low uncertainty because the transient models do not

violate any assumptions underlying $\chi$-based methods. The spatially variable model runs, where assumptions of the $\chi$ method are violated, still perform better than slope–area analysis in extracting the correct $m/n$ ratio. This gives us some confidence that in real landscapes, where non-uniform uplift and spatially varying erodibility are likely pervasive, extracted $m/n$ ratios may still reveal useful information about the incision processes. In addition, $\chi$ profiles can be used to infer whether most likely $m/n$ ratios vary greatly between basins due to heterogeneous channel profiles, which could be caused by variable erodibility or

tectonics, or if tributaries are well aligned and the variability in $m/n$ ratio may be due to an underlying pattern in the mechanics of channel incision. We present results from some real landscapes to highlight possible scenarios that will be encountered by users of our methods, and to suggest potential areas for future research.

*Code and data availability.* Code used for analysis is located in the LSDTopoTools github repository: https://github.com/LSDtopotools/ LSDTopoTools_ChiMudd2014, and scripts for visualising the results can be found at https://github.com/LSDtopotools/LSDMappingTools.

We have also provided documentation detailing how to install and run the software which can be found at https://lsdtopotools.github.io/ LSDTT_documentation. As part of the supplementary information we have also provided example parameter files which can be used to reproduce the results of all analyses performed in this study.

*Author contributions.* SMM and FJC wrote the code for the analysis, performed the numerical modelling and wrote the visualisation software. SMM, FJC, BG, and MDH performed the analysis on the real landscapes. SMM wrote the paper with contributions from other authors.

*Acknowledgements.* We thank Rahul Devrani, Jiun Yee Yen, Ben Melosh, and Julien Babault for beta testing the software. This work was supported by Natural Environment Research Council grants NE/J009970/1 to Mudd, NE/P012922/1 to Clubb. Gailleton was funded by European Union Initial Training Grant 674899 — SUBITOP.





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





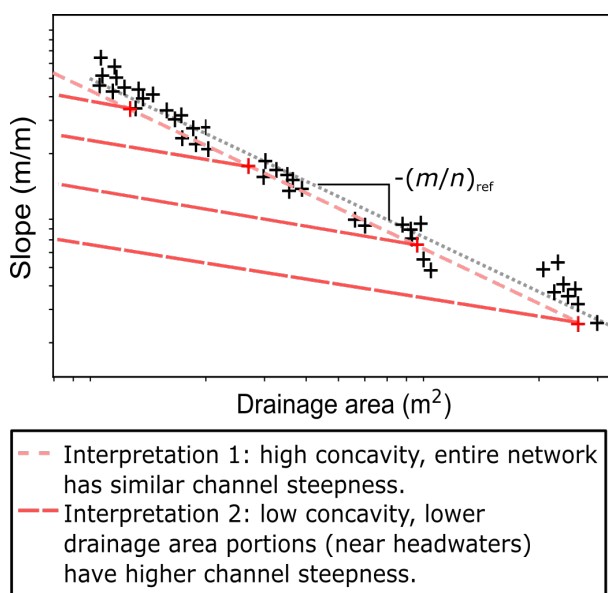

**Figure 1.** Sketch illustrating the effect of choosing different reference $m/n$ ratios. A simple regression of the data suggests that all parts of the channel network have similar values of $k_{sn}$. However, if a lower reference $m/n$ ratio is chosen, the $k_{sn}$ values will be systemically higher for channels at lower drainage area. This sketch is based on data from a numerical simulation where the latter situation has been imposed via higher uplift rates in the core of the mountain range, showing the potential for incorrect $m/n$ ratios to be extracted from slope–area data alone.





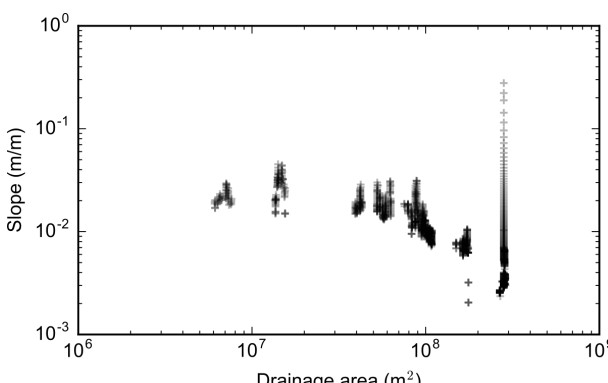

**Figure 2.** A typical slope–area plot. This example is from a basin near Xi'an, China, with an outlet at approximately 34°26'23.9"N 109°23'13.4"E. The slope–area data typically contains gaps due to tributary junctions, as well as wide ranges in slope for the reaches between junctions due to topographic noise inherent in deriving slope values. The result is a high degree of scatter in the data. These data are produced by averaging slope values over a fixed vertical interval of 20 m.





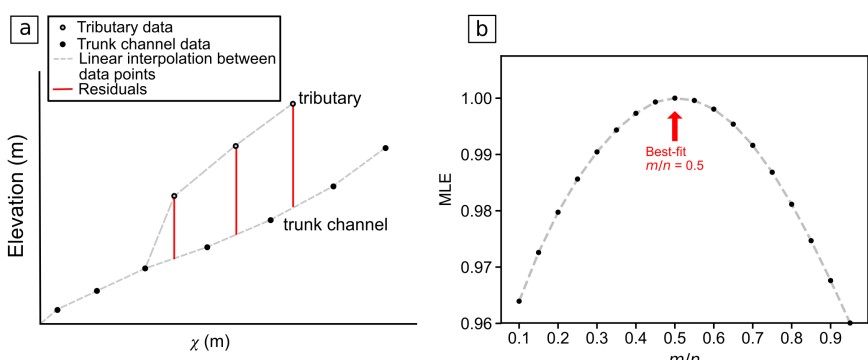

**Figure 3.** Sketch illustrating the methodology of the $\chi$ method using all profile data, where (**a.**) residuals between tributary and trunk channel $\chi$–elevation data are calculated by using linear fits between data on the trunk channel, and (**b.**) the variation in the maximum likelihood estimator (MLE), calculated using equation (11), is used to select the most likely $m/n$ ratio.




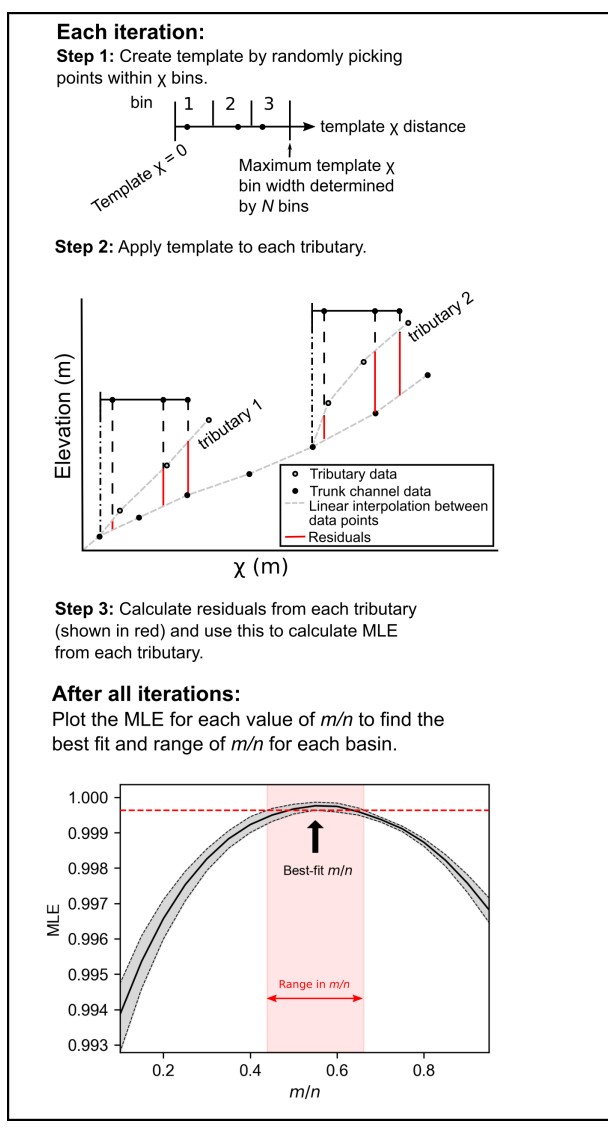

**Figure 4.** Sketch showing how we compute residuals for our Monte-Carlo points method of determining the maximum likelihood estimator (MLE) of the $m/n$ ratio, and then use the uncertainty in MLE values to compute the uncertainty in the $m/n$ ratio.





**Figure 5.** Shaded relief plots of the model runs with temporally varying uplift, with drainage basins plotted by the best fit $m/n$ predicted from the $\chi$ Monte Carlo analysis (first column), and slope-area analysis (second column). The basins are coloured by the predicted $m/n$ ratio, where darker colours indicate a higher $m/n$. The extracted channel network for each basin is shown in blue.





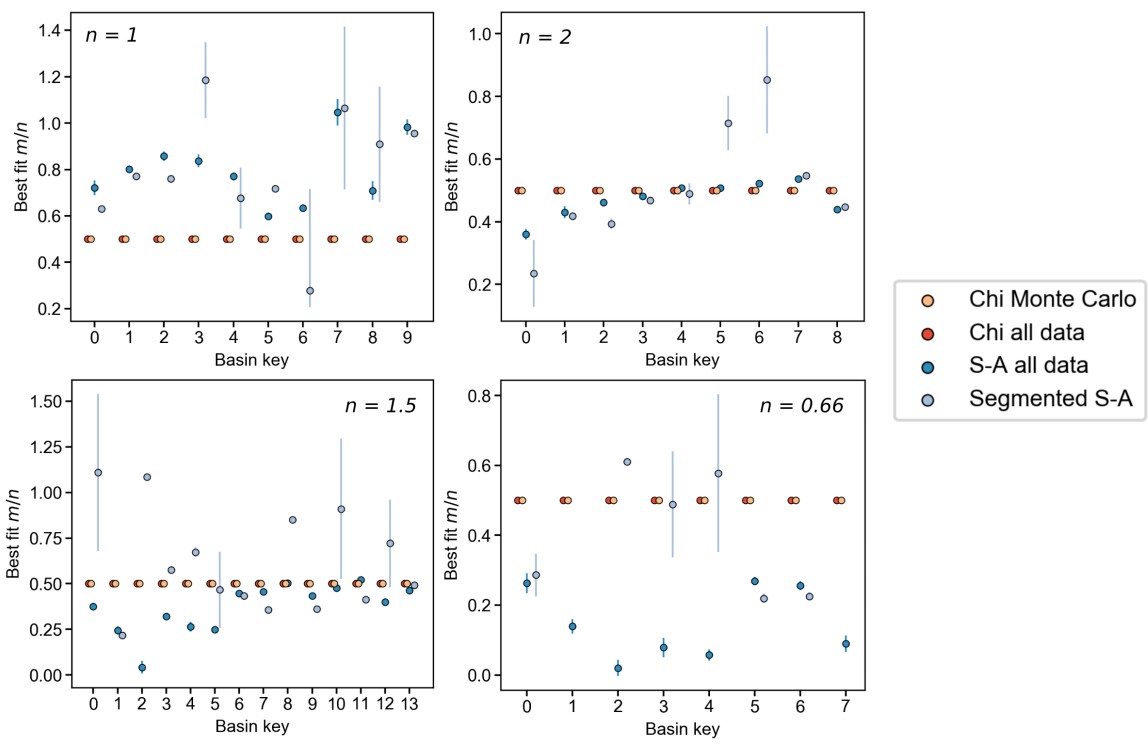

**Figure 6.** Plots showing the predicted best fit $m/n$ ratio for each basin and each method for $m/n = 0.5$, where $n = 1$, $n = 2$, $n = 1.5$, and $n = 0.66$. The $\chi$ methods are shown in reds and the slope-area methods are shown in blues.



**Figure 7.** Results of the model runs with spatially varying erodibility ($K$, left column) and uplift ($U$, right column). The top four panels show the spatial pattern of predicted $m/n$ from the $\chi$ Monte Carlo analysis and the slope-area analysis, where the basins are coloured by $m/n$ (darker colours = higher $m/n$). The bottom two panels show density plots of the distribution of $m/n$ for each method, where the dashed line marks the correct $m/n = 0.5$.





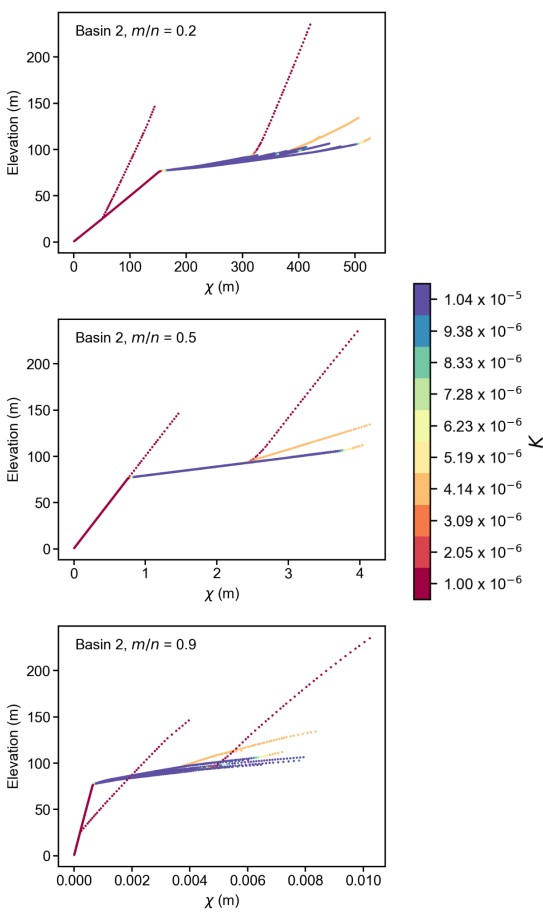

**Figure 8.** Example $\chi$-elevation plots for the model run with spatially varying erodibility, where points are coloured by $K$. The $m/n$ increases in each plot from 0.2 to 0.9. Tributaries with the same $K$ value are collinear in $\chi$-elevation space.



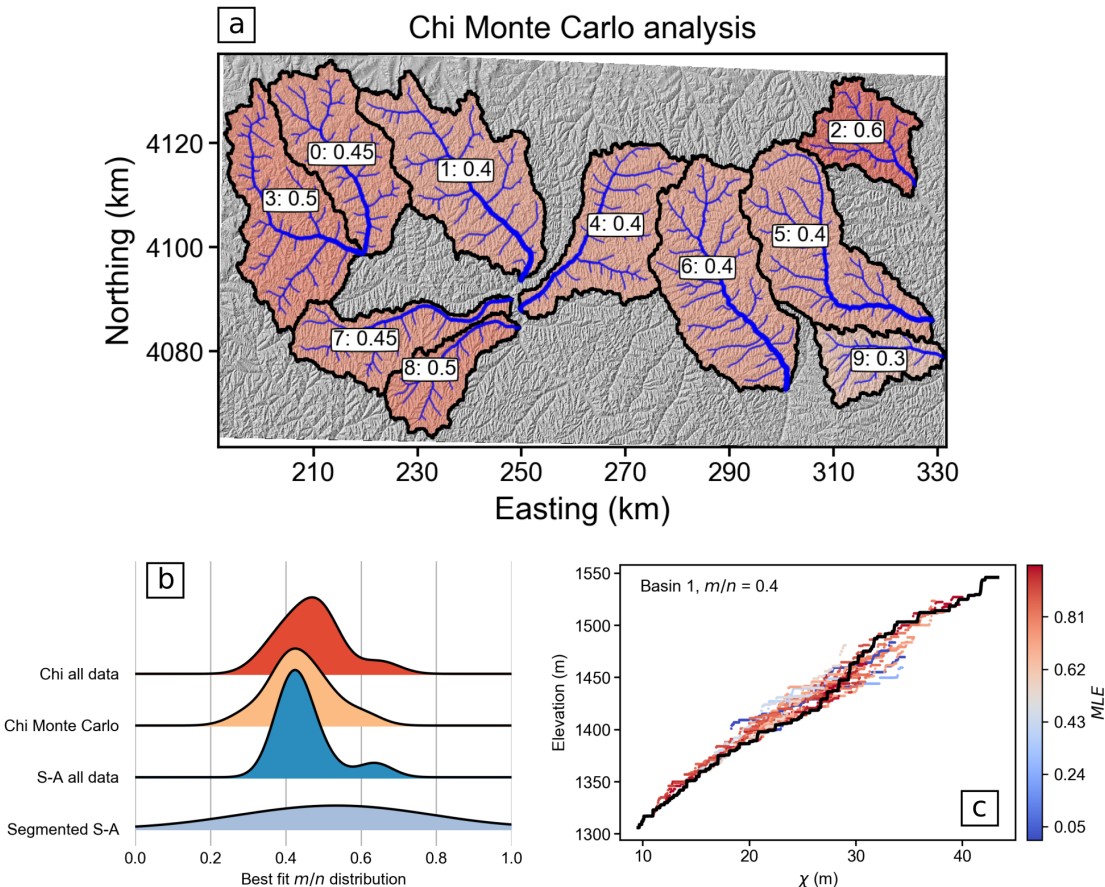

**Figure 9.** Exploration of the most likely $m/n$ ratio in the Loess Plateau, China, UTM Zone 49°N. Basins with the most likely $m/n$ ratio determined by the Monte Carlo points method is displayed in panel **a.**; the basin number is followed by the most likely $m/n$ in the basin labels. The probability density of best fit $m/n$ ratio using the individual basins' most likely $m/n$ is shown in panel **b.**. The $\chi$–elevation plot for the most likely $m/n$ in basin 1 determined from the two $\chi$ methods shown in panel **c.**



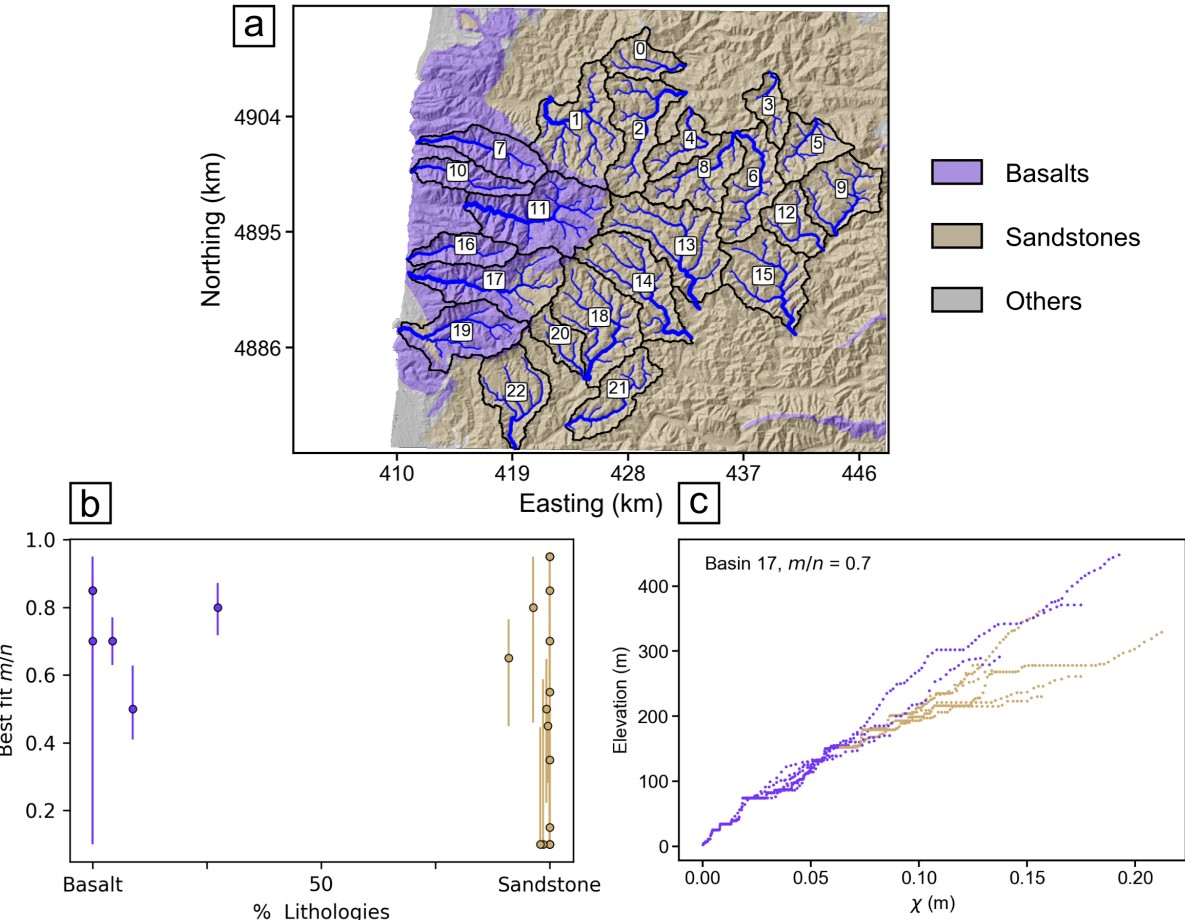

**Figure 10.** Exploration of the most likely $m/n$ ratio near Waldport, Oregon, UTM Zone 10°N. Basins numbers and the underlying lithology is displayed in panel **a.**. The most likely $m/n$ ratio determined by the Monte Carlo points method as a function of the percent of each basin in the different lithologies is shown in panel **b.** Panel **c.** shows the $\chi$–elevation plot for a basin that has two bedrock types; the channel pixels are coloured by lithology. The plot uses the typical $m/n$ ratio for basalt (0.7).





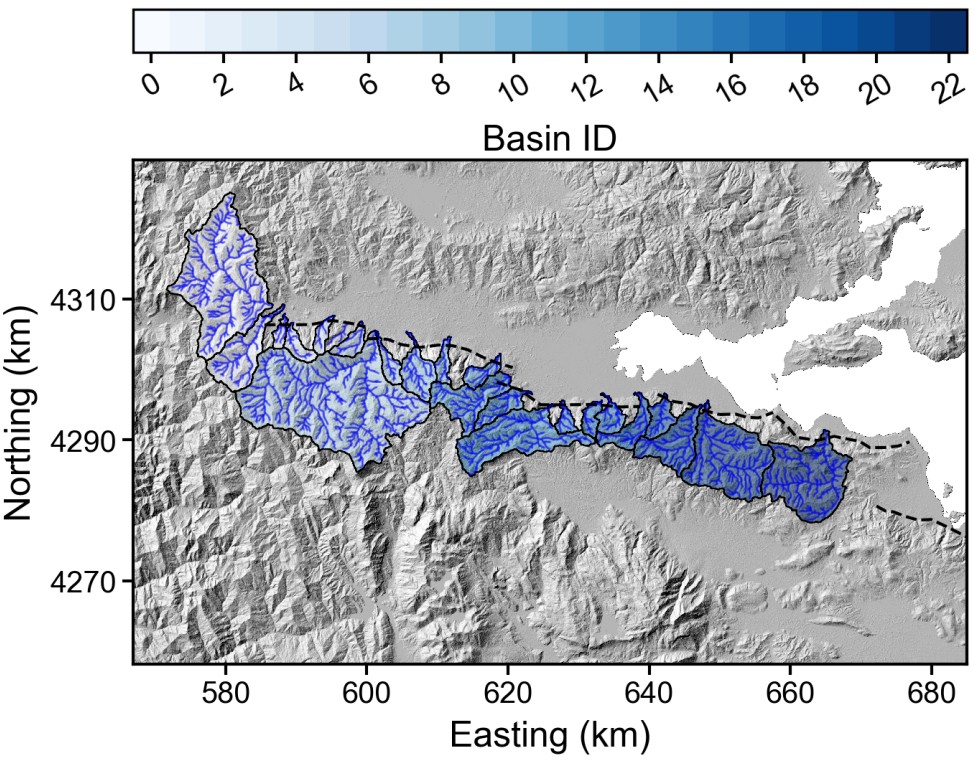

**Figure 11.** Basins analysed near the Gulf of Evia, Greece, UTM Zone 34°N that interact with active normal faults previously studied by Whittaker and Walker (2015).



Earth **Surface**
**Dynamics**
Discussions

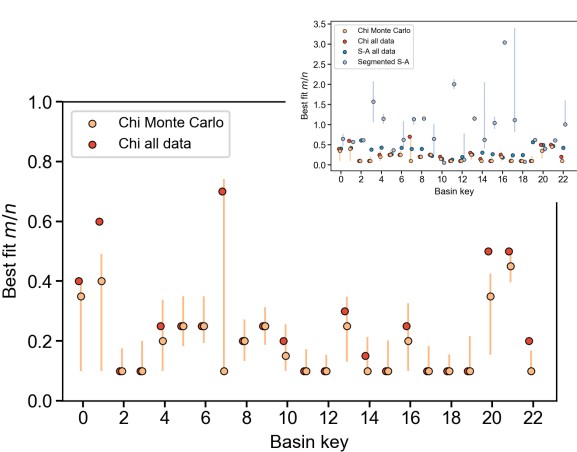

**Figure 12.** The predicted best fit $m/n$ ratio determined using the $\chi$ methods (red points) and slope-area methods (blue points shown in inset).
Basin numbers correspond to those plotted in Figure 11.





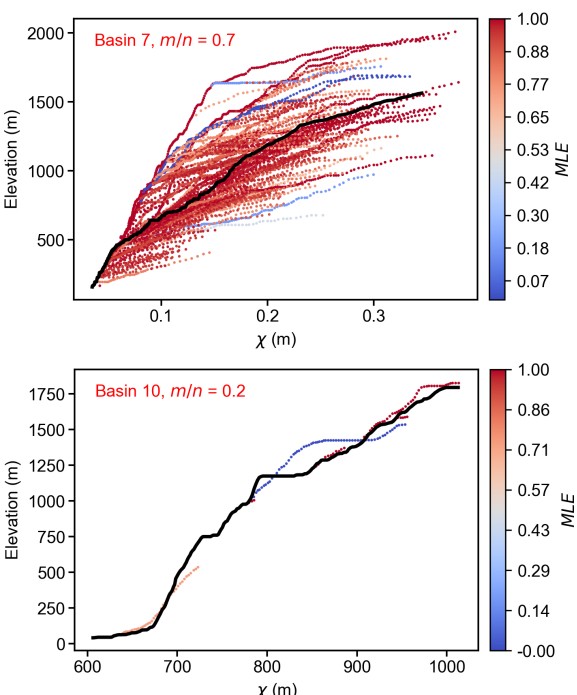

**Figure 13.** Profile $\chi$–elevation plots associated with best fit $m/n$ ratio for Basin 7, a large catchment with many tributaries draining across a relay zone between normal fault segments, and Basin 10, a small, steep catchment draining directly across the footwall segment of a normal fault with few tributaries.

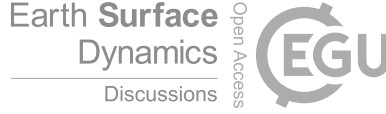



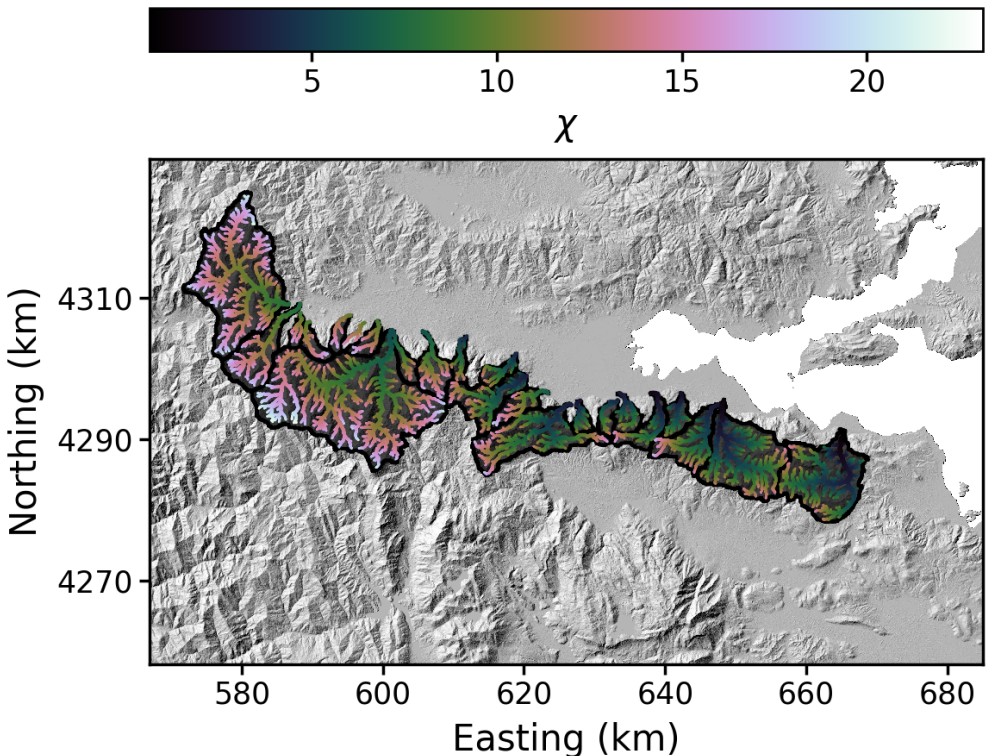

**Figure 14.** Spatial distribution of the $\chi$ coordinate in the channel network calculated using $A_0 = 1$; $m/n = 0.45$. Gradients in $\chi$ across topographic divides (black) can indicate planform disequilibrium such that the drainage network may be reorganising. Divides will tend to migrate from low values of $\chi$ towards high values in the channel network.