# Peer review of "How concave are river channels?"

_Earth Surface Dynamics, 2018_

## Referee Comment (RC1) · R. DiBiase (Referee) · 27 Feb 2018

This paper presents a new method for constraining the intrinsic concavity of river channels, in order to more accurately interpret spatiotemporal patterns of climate and tectonics from landscapes that deviate from the simpler case of steady state, uniform rock uplift, rock strength, and climate. The new metric compares the chi-elevation plots of tributary and mainstem channels in an objective manner, and is integrated into LSDTopoTools, an open source topographic analysis environment developed by the authors. This paper then evaluates the model as compared to existing approaches, using examples from real and synthetic landscapes.

Overall, this is a nicely-written paper with great figures and the code seems like a very useful addition to an arsenal of topographic analysis scripts that have evolved in recent years (e.g., LSDTopoTools and TopoToolbox). I think this paper fits well at ESurf, and I

only have one major issue that I think needs to be resolved before publication:

Major comment: On Page 4, Line 25, the authors recognize a strength of the existing slope-area method of determining channel concavity is that it "requires no assumptions whatsoever about the underlying form of the equations describing channel incision". Thus, I was surprised to find that the chi analysis underpinning the new method was (unnecessarily) framed in terms of the stream power model! Although the Perron and Royden 2012 paper also frames chi in terms of stream power, I would instead recast equations 7-9 in terms of the more general empirical relationship of Flint's law (equation 1), which makes no assumptions about process – ks and theta are simply geometrical properties of river channels. We did this in Whipple et al. 2017 Geology (doi:10.1130/G38490.1), but did not expand too much on the reasoning.

Note also that the relationship between channel steepness and erosion rate/uplift rate (Page 3, Line 21-29) is again not necessarily tied to stream power, but relates to an empirical relationship between relief and erosion rate (equation 1 of DiBiase and Whipple, 2011, doi:10.1029/2011JF002095; also discussed in Whipple and Meade 2006, doi:10.1016/j.epsl.2005.12.022). Connecting this exponent and the concavity index to m's and n's in stream power gets problematic because things vary depending on the specific form of the incision law (for example, adding a threshold changes the steepness-E relationship without changing m or n).

I think the paper would be stronger if, like the title says, the main analysis focuses on finding the intrinsic concavity index theta, rather than the model-dependent ratio m/n. Note that this of course does not preclude the comparison with stream power model landscapes shown in section 3 and interpretation/comparison with expected m/n!

————— Line comments: —————

Page 5, Line 16: I think only the profile is smoothed, rather than the full DEM.

Page 5, Line 23: Is method (i) using a single channel, the entire channel network?

Whole DEM?

Page 7, Line 1: This is just one new method, correct?

Page 7, Line 9-10: Not totally necessary, but might be helpful to emphasize the MLE = 1 for r = 0.

Page 7, Line 11: There seems to be a mistake in the math here where it was assumed that exp(ab) = exp(a)exp(b) rather than exp(a)^b.

Page 8, Line 5-9: Not just hanging tributaries, but any complexities influencing concavity that are not captured by simple stream power framework (e.g., spatial patterns in sediment cover/grain size). Perhaps it makes sense to include areas upstream of these hanging tributaries in the statistical analysis? Maybe collinearity is too stringent, and similar steepness is instead more useful?

Page 9, Line 34: "i) by regression of all $\chi$-elevation data" Make clear whether this is just one channel or the whole tributary network at once

Page 12, Line 3-10: Typo—this text is directly repeated from above.

Page 12, Line 32: Note that Duvall et al. (2004) argue that the high concavities in the Santa Ynez Mtns are due to strong rocks in the headwaters and weak rocks below, which is different than the "spatially varying m/n as a function of lithology" shown in Fig. 10.

Page 13, Line 19-20: I agree - but then why is it appropriate to use this for the numerical experiment on landscape transience, which also includes knickpoints?

Page 13, Line 23: I think more importantly, other processes become important in the transient! (e.g., DiBiase et al, 2015, doi:10.1130/B31113.1)

Page 14, Line 7: Spatial gradients in tectonics are far more important than temporal variations in disrupting interpretations of chi at divides. if spatially uniform U/K, then chi still good indicator of divide instability during temporally varying U (or K).

[Figure]

Page 14, Line 9-18: I don't quite agree here. The fact that this is a relay system means that spatially variably uplift likely dominates, complicating a simple interpretation of chi across divides (see Whipple et al., 2017 JGR, doi:10.1002/2016JF003973)

Page 14, Line 19-22: "...river profiles...are not alone sufficient to interpret the history of landscape evolution, but must be considered alongside other observational data and in the context of a process-based understanding of landscape evolution ..." I strongly agree!

Page 14, Line 21: Typo "bust"

Page 14, Line 32: Be careful tying the paper to stream power! (see main comment above)

Page 15, Line 4-6: I think would be good to point out that the second method does not handle well spatially variable rock uplift rate.

Figure 1: More detail is needed in caption to explain this sketch. Is it a single trunk channel? An entire stream network? There is also some good discussion of these challenges of interpreting concavity in Gasparini and Whipple, 2014 (doi:10.1130/L322.1).

Figure 2: Again, is this a single channel? Whole tributary network?

Figure 3: This caption could use more description. Hard to follow without careful reading of main text.

Figure 11: Do you mean "UTM Zone 34N"?

---

## Referee Comment (RC2) · L. Goren (Referee) · 16 Mar 2018

The manuscript presents and compares several techniques for extracting the concavity index of fluvial basins from topographic fluvial data. The manuscript nicely states how, for different (yet, specific) models of fluvial incision, the true, process-dependent (or process-assumed), concavity index is a crucial parameter, without which, the steepness index and information about time and space dependent uplift rates cannot be reliably retrieved. The importance of the concavity index and the motivation behind the presented analyses are therefore convincing.

The manuscript is well written, and the effort that was invested in articulating the scope of the problem and the different techniques and analyses eases the reading of even complicated concepts.

Overall, the manuscript compares between two classes of techniques for extracting

the concavity index, slope-area analysis and chi-z analysis. Through several insightful numerical examples the superiority of the chi-z analysis is demonstrated in particular for spatially heterogeneous and transient landscapes. The manuscript then turns to explore the concavity index of natural landscapes, where the conclusions are, as expected, more ambiguous.

I have one major concern: Given that the manuscript is methodological in nature, namely, it explores the accuracy and robustness of different techniques for evaluating the concavity index, it is lacking essential reasoning for developing a new technique without exploring existing ones or even just pointing out their possible theoretical limitations. Here, I specifically refer to the development of the maximum likelihood estimator for m/n from chi analysis (which is split into two techniques), without exploring existing techniques such as the 'tributary scatter reduction' (Goren et al., 2014) and a later version of this technique developed in Hergarten et al., 2016 (both papers are cited in the manuscript). These techniques find the m/n that minimizes the scatter in elevation over chi bins. They are intuitive, computationally simple, and the scatter itself can be used to evaluate the uncertainty. Developing a new technique that appears to be computationally more demanding without comparing and contrasting it to existing techniques does not serve the goals of the manuscript and of the community that can benefit from it.

On the same note, I would like to draw the authors attention to a pre-print https://eartharxiv.org/5u9eg/ (recently accepted for publication in JGR-ES) that, for a different geomorphic application, compares m/n values derived from slope-area and from chi-z using the tributary scatter reduction technique. I'm a co-author on this manuscript and I apologize for this far from elegant self-promotion, but it's very relevant to the current manuscript under discussion.

Another, more minor, comment, is that currently, the manuscript is missing a discussion about which and under what conditions each of the two chi-based techniques for extracting m/n is better.

Additional comments:

Page 3, line 4: Within the scope of the current manuscript the adjective 'constant' for m and n is a bit misleading.

Page 6, line 9: 'The chi coordinate is simply a derived function of topography'. It's a function of the distribution of the drainage area, or the topology, and not of the topography.

Page 7, lines 15-17: The technique of minimizing z scatter over chi bins that was mentioned above does not have this issue.

Page 7, lines 22: Could it be that 'bootstrapping' is a more accurate description than 'Monte-Carlo'?

Page 8, line 13: 'must'

Page 10, line 19: The geometry of the K patches should be described. From the fig, they appear to be square-shaped. Wouldn't it make more sense for the patches to be a function of the topography of even the drainage network itself?

Page 12, line 3: 'reference concavities between 0.4 and 0.5 should give an accurate representation of the relative steepness'. Do you mean that in general or just for the Loess Plateau? If generally, then it calls for a justification. How does it relate to your natural basalt-sandstone experiment in Oregon?

Page 12, lines 3-10: repeated text.

Page 13, line 2: A short discussion of how the lithology is expected to affect m/n is probably needed here. (Possibly via the relation between channel width and specific stream power/drainage area?)

Page 13, lines 15-16: Could be worth mentioning that the Gulf of Evia overall represents a natural experiment where U varies both temporally and spatially.

Page 14, lines 15-18: How exactly does drainage area change affect the derived m/n? If all the tributaries are losing area, then they should all be plotted as convex in the chi-z domain. But the technique tries to minimize the residual and not to straighten the profiles. How is the residual affected by area change?

Page 14, line 21: 'bust'

Page 15, lines 13-16: This appears to be a key sentence, but its relation to the results and discussion is not straightforward.

Fig 7: maybe it's worthwhile explaining what are the squared low relief patches in the variable K panels.

Fig 9: The captions of panel C are not clear. The two chi-based methods have different m/n maxs.

Fig 11: I assume that the dashed line represents faults. Maybe add a legend. Also, it might be worth differentiating (by color) between basins that drain across relay ramps and those that drain across faults.

Fig 12: Same comment: differentiate between basins that drain across relay ramps and those that drain across faults.

Fig 13: From my experience in chi-z analysis, such a scatter and concave tributaries are indicative that the chosen m/n is too high. Can you show the same basin with different m/n. This might hint that the scatter minimization technique and your new MLE technique give different results.

Wang 2017b probably deserves more credit for comparing the chi-z to slope-area predictions.

---

## Editor Comment (EC1) · JM Turowski (Editor) · 23 Mar 2018

Dear authors,

we have now received two generally positive reviews of the paper. The issues raised by the reviewers seem self-explanatory and fairly straight-forward to deal with, and I do not think that I need to elaborate on them. I would just like to highlight a small comment by reviewer #1: there currently is a slight mismatch between the title and the content of the paper. By just reading the title, the reader may not expect a methodological paper, and the question currently posed in the title is not actually adressed in the paper. I encourage you to re-think the title such that it better reflects the content and aims of the paper.

Best, Jens

[Figure]

**ESurfD**

---

## Author Comment (AC1) · 21 Apr 2018

We thank reviewer 1 (Roman DiBiase) for his thorough review and for highlighting a different way of casting the paper that does not rely on stream power. We will still make some mention of stream power because it serves as the basis for numerical simulations, and also plays a role in the assumption of collinearity (see below), but we take the advice that introducing the concept of concavity can be done without this restrictive assumption. These comments have very much helped make the context of the paper more general, and we would like to thank the reviewer for these suggestions which we feel have substantially improved the paper.

*This paper presents a new method for constraining the intrinsic concavity of river channels, in order to more accurately interpret spatiotemporal patterns of climate and tectonics from landscapes that deviate from the simpler case of steady*

[Figure]

*state, uniform rock uplift, rock strength, and climate. The new metric compares the chi-elevation plots of tributary and mainstem channels in an objective manner, and is integrated into LSDTopoTools, an open source topographic analysis environment developed by the authors. This paper then evaluates the model as compared to existing approaches, using examples from real and synthetic landscapes. Overall, this is a nicely-written paper with great figures and the code seems like a very useful addition to an arsenal of topographic analysis scripts that have evolved in recent years (e.g., LSDTopoTools and TopoToolbox). I think this paper fits well at ESurf, and I only have one major issue that I think needs to be resolved before publication:*

Thank you for your supportive comments. As we describe below, we agree with the suggested revision (see below) and will carry it out in the revised manuscript.

*Major comment: On Page 4, Line 25, the authors recognize a strength of the existing slope-area method of determining channel concavity is that it "requires no assumptions whatsoever about the underlying form of the equations describing channel incision". Thus, I was surprised to find that the chi analysis underpinning the new method was (unnecessarily) framed in terms of the stream power model! Although the Perron and Royden 2012 paper also frames chi in terms of stream power, I would instead recast equations 7-9 in terms of the more general empirical relationship of Flint's law (equation 1), which makes no assumptions about process – ks and theta are simply geometrical properties of river channels. We did this in Whipple et al. 2017 Geology (doi:10.1130/G38490.1), but did not expand too much on the reasoning.*

Yes, we agree this is a much better approach. We have modified the text accordingly and now take the approach outlined in the supplementary materials of the Whipple et al paper. The introduction now separates concavity from stream power. However we cannot escape stream power entirely. In chi space, if we only rely on concavity then the prediction is the chi profiles are linear. But any disturbance in

erosion rates will lead to piecewise linear segments (Royden and Perron (2013) showed this very nicely). In the Mudd et al (2014) paper, we attempted to address this using a segmentation algorithm. However, this algorithm is sensitive to parameter values as well as being computationally expensive. By running many hundreds of landscapes through our code since then, we found that tests of collinearity appeared to perform better than assuming any linearity to segments along the profile. Perron and Royden (2013) suggested that if erosion signals propagated upward in elevation at the same rate, then the elevations at the same chi distance along the channel will be the same. In other words, the tributaries and trunk channel will be collinear. We therefore decided to run this test in idealised situations, where we could exactly constrain the concavity via $m/n$ in the stream power incision model. The problem with the collinearity test is that it cannot be derived from Flint's law: it rests upon assumptions about how erosion signals will move through the channel network. Stream power is one theory that predicts this behaviour. Therefore, the collinearity test has to be connected to an incision law and not directly to Flint's law, meaning that we cannot divide our analysis from stream power completely. This is somewhat unsatisfying, as we are well aware of the many assumptions that go into the stream power model (e.g. Lague (2014)). We have completely rewritten the introduction to address this point and make clear the assumptions involved.

*Note also that the relationship between channel steepness and erosion rate/uplift rate (Page 3, Line 21-29) is again not necessarily tied to stream power, but relates to an empirical relationship between relief and erosion rate (equation 1 of DiBiase and Whipple, 2011, doi:10.1029/2011JF002095; also discussed in Whipple and Meade 2006, doi:10.1016/j.epsl.2005.12.022). Connecting this exponent and the concavity index to m's and n's in stream power gets problematic because things vary depending on the specific form of the incision law (for example, adding a threshold changes the steepness-E relationship without changing m or n).*

We will highlight the papers mentioned here and ensure that the connection between steepness and erosion rates are clear, regardless of any assumed incision law.

*I think the paper would be stronger if, like the title says, the main analysis focuses on finding the intrinsic concavity index theta, rather than the model-dependent ratio m/n. Note that this of course does not preclude the comparison with stream power model landscapes shown in section 3 and interpretation/comparison with expected m/n!*

Duly noted. We will change the focus to the actual concavity rather than the *m/n* ratio so the title is now more indicative of the paper contents.

*Page 5, Line 16: I think only the profile is smoothed, rather than the full DEM.*

The Wobus paper actually recommends smoothing the DEM: it was written in the dark ages of DEM quality. However we will note that modern workers don't do this.

*Page 5, Line 23: Is method (i) using a single channel, the entire channel network? Whole DEM?*

We will clarify this in the text: it uses all the tributaries and the main stem in a given basin.

*Page 7, Line 1: This is just one new method, correct?*

We now call it two (there is the all points and what we were calling the 'monte carlo points' methods). Liran Goren suggested we call the second method a bootstrap method and we will use this terminology. We also introduce a disorder metric suggested by Liran and will devise a version of this to account for uncertainty.

*Page 7, Line 9-10: Not totally necessary, but might be helpful to emphasize the MLE = 1 for r = 0.*

Done

*Page 7, Line 11: There seems to be a mistake in the math here where it was assumed that exp(ab) = exp(a)exp(b) rather than exp(a)ËĘb.*

Thanks for spotting that. We inserted this mistake as a rhetorical device and it doesn't affect the results. We will expunge this equation from the manuscript.

*Page 8, Line 5-9: Not just hanging tributaries, but any complexities influencing concavity that are not captured by simple stream power framework (e.g., spatial patterns in sediment cover/grain size). Perhaps it makes sense to include areas upstream of these hanging tributaries in the statistical analysis? Maybe collinearity is too stringent, and similar steepness is instead more useful?*

See discussion early in the response. We acknowledge the problems with SPIM and the fact that the collinearity method relies on assumptions about process. However a local linearity test requires some segmentation process (which is what some of the authors of this paper tried in 2014 and we find that method is extremely noisy and uncertain). We have tried to highlight the drawbacks of collinearity but we feel its advantages outweigh its disadvantages. We now say this in the conclusion, and explain why we feel this way.

*Page 9, Line 34: "i) by regression of all $\chi$-elevation data" Make clear whether this is just one channel or the whole tributary network at once*

We will now say "For all but the final method the analyses use all tributaries in the basins."

*Page 12, Line 3-10: Typo: This text is directly repeated from above.*

Fixed.

*Page 12, Line 32: Note that Duvall et al. (2004) argue that the high concavities in the Santa Ynez Mtns are due to strong rocks in the headwaters and weak rocks below, which is different than the "spatially varying m/n as a function of lithology" shown in Fig. 10.*

We will modify the text so as not to mislead readers.

*Page 13, Line 19-20: I agree - but then why is it appropriate to use this for the numerical experiment on landscape transience, which also includes knickpoints?*

In practice, workers generally fit small sections of the channel network with a concavity because the knickpoints distort the overall concavity. This is typically done using visual inspection. The tutorials and code associated with the Wobus et al. (2006) paper, for example, include functions to let users manually choose intervals over which to select concavity. One of our main goals is to ensure reproducibility, so we attempted to use a segment finding algorithm (mentioned in the paper). This sometimes works, and sometimes doesn't. So we find it very difficult to select appropriate segments for concavity using S-A data with reproducible techniques–this is true for both numerical simulations as well as in real landscapes. I suppose we are setting up a straw man for the numerical simulations, but this straw man situation is the one every geomorphologist finds themselves in when they are doing S-A analysis.

*Page 13, Line 23: I think more importantly, other processes become important in the transient! (e.g., DiBiase et al, 2015, doi:10.1130/B31113.1)*

Indeed. We will add a sentence here highlighting this. Although, many of these other processes (i.e., debris flows) tend to reduce concavity and not increase it.

*Page 14, Line 7: Spatial gradients in tectonics are far more important than temporal variations in disrupting interpretations of chi at divides. if spatially uniform U/K, then chi still good indicator of divide instability during temporally varying U (or K).*

We will note this in the revision.

*Page 14, Line 9-18: I don't quite agree here. The fact that this is a relay system means that spatially variably uplift likely dominates, complicating a simple interpretation of chi across divides (see Whipple et al., 2017 JGR, doi:10.1002/2016JF003973)*

We introduced this example simply to show that things could get complicated so we have tried to avoid too much interpretation (since we are not really in a position to do so). However we will add a sentence to make the point highlighted by the reviewer. The other reviewer has also asked for some adjustments here so there will be some changes to this section.

*Page 14, Line 19-22: "...river profiles...are not alone sufficient to interpret the history of landscape evolution, but must be considered alongside other observational data and in the context of a process-based understanding of landscape evolution ..." I strongly agree!*

We are optimistic that a richer set of metrics will be used in the future as topographic and other data improves.

**ESurfD**
*Page 14, Line 21: Typo "bust"*

Fixed.

*Page 14, Line 32: Be careful tying the paper to stream power! (see main comment above)*

See discussion above. We completely agree with the reviewer about having metrics that are agnostic toward channel incision laws but the collinearity test derives from the SPIM so can't be described as purely geometric. We acknowledge this is a weakness of the test.

*Page 15, Line 4-6: I think would be good to point out that the second method does not handle well spatially variable rock uplift rate.*

We will rewrite this somewhat to point this out and also introduce the results from the new disorder test (which in some cases does better).

*Figure 1: More detail is needed in caption to explain this sketch. Is it a single trunk channel? An entire stream network? There is also some good discussion of these challenges of interpreting concavity in Gasparini and Whipple, 2014 (doi:10.1130/L322.1).*

We will clarify this in the caption. In the text around the reference to the figure we will refer to the very nice Gasparini and Whipple (2014) paper.

*Figure 2: Again, is this a single channel? Whole tributary network?*

Clarified. We now say "The data is taken from only the trunk channel."

*Figure 3: This caption could use more description. Hard to follow without care-ful reading of main text.*

We will expand on the caption.

*Figure 11: Do you mean "UTM Zone 34N"?*

Yes we do. Thanks for finding that. It will be fixed.

---

## Author Comment (AC2) · 21 Apr 2018

We thank reviewer 2 (Liran Goren) for a number of insightful comments that will help improve the paper. Reviewer 2 is entirely correct that we should have tested the disorder metric that has been used in several recent papers. We have followed this advice and we find that it performs similarly to the "all chi" method and in some cases is the most accurate method. It is also more computationally efficient than other methods. The only real drawback is that because it uses all data it cannot express the uncertainty in the concavity, so we are now in the process of designing an algorithm to estimate its uncertainty.

*The manuscript presents and compares several techniques for extracting the concavity index of fluvial basins from topographic fluvial data. The manuscript nicely states how, for different (yet, specific) models of fluvial incision, the true, process-*

[Figure]

*dependent (or process-assumed), concavity index is a crucial parameter, without which, the steepness index and information about time and space dependent uplift rates cannot be reliably retrieved. The importance of the concavity index and the motivation behind the presented analyses are therefore convincing.*
*The manuscript is well written, and the effort that was invested in articulating the scope of the problem and the different techniques and analyses eases the reading of even complicated concepts.*

Thanks. We are glad to hear that the manuscript is clear.

*Overall, the manuscript compares between two classes of techniques for extracting the concavity index, slope-area analysis and chi-z analysis. Through several insightful numerical examples the superiority of the chi-z analysis is demonstrated in particular for spatially heterogeneous and transient landscapes. The manuscript then turns to explore the concavity index of natural landscapes, where the conclusions are, as expected, more ambiguous.*

Natural landscapes are indeed a vexing problem since there is no way to know the "real" concavity or $m/n$ ratio, but hopefully our contribution at least allows others to estimate concavity and $m/n$ reproducibly.

*I have one major concern: Given that the manuscript is methodological in nature, namely, it explores the accuracy and robustness of different techniques for evaluating the concavity index, it is lacking essential reasoning for developing a new technique without exploring existing ones or even just pointing out their possible theoretical limitations. Here, I specifically refer to the development of the maximum likelihood estimator for m/n from chi analysis (which is split into two techniques), without exploring existing techniques such as the 'tributary scatter reduction' (Goren et al., 2014) and a later version of this technique developed in Hergarten et al.,*

*2016 (both papers are cited in the manuscript). These techniques find the m/n that minimizes the scatter in elevation over chi bins. They are intuitive, computationally simple, and the scatter itself can be used to evaluate the uncertainty. Developing a new technique that appears to be computationally more demanding without comparing and contrasting it to existing techniques does not serve the goals of the manuscript and of the community that can benefit from it.*

We agree, this was an oversight. We have now implemented the disorder metric and tested it against our simulated landscapes. It does quite well! Please see Figures 1 and 2 (at the end of this document).

These are not final figures as we are still developing an algorithm to estimate the uncertainty of this method. Because the disorder metric uses an ordered list of elevations, and the ordering affects the disorder metric, we cannot remove individual points as we did in the Monte Carlo approach in the discussion paper (we will call this the bootstrap method based on advice in this review). At the moment we are trying all combinations of tributaries to enter into the disorder metric. The main challenge is that we are using different methods to quantify uncertainty so this could mislead others. We can't see a common method across all metrics to quantify uncertainty so we are going to emphasise in the text that these uncertainty metrics are only qualitative.

*On the same note, I would like to draw the authors attention to a pre-print https://eartharxiv.org/5u9eg/ (recently accepted for publication in JGR-ES) that, for a different geomorphic application, compares m/n values derived from slope-area and from chi-z using the tributary scatter reduction technique. I'm a co-author on this manuscript and I apologize for this far from elegant self-promotion, but it's very relevant to the current manuscript under discussion.*

The paper is very interesting and we are happy to have it brought to our attention. The findings in that study are relevant to our work and will cite it in the paper.

*Another, more minor, comment, is that currently, the manuscript is missing a discussion about which and under what conditions each of the two chi-based techniques for extracting m/n is better.*

We now include such a discussion.

*Page 3, line 4: Within the scope of the current manuscript the adjective 'constant' for m and n is a bit misleading.*

Deleted the word "constant".

*Page 6, line 9: 'The chi coordinate is simply a derived function of topography'. It's a function of the distribution of the drainage area, or the topology, and not of the topography.*

This sentence no longer appears since we have reorganised how we introduce chi by integrating Flint's law, as requested by the first reviewer. But thanks for pointing this out because we probably would have said it in another paper.

*Page 7, lines 15-17: The technique of minimizing z scatter over chi bins that was mentioned above does not have this issue.*

Well, the disorder statistic will still have this issue because longer tributaries will diverge from the trunk channel more so will add more weight to the disorder statistic. But we now implement the disorder statistic for the paper and have a discussion of its relative strengths (see above).

*Page 7, lines 22: Could it be that 'bootstrapping' is a more accurate description than 'Monte-Carlo'?*

Yes, you are correct. We have changed the name.

*Page 8, line 13: 'must'*

Fixed.

*Page 10, line 19: The geometry of the K patches should be described. From the fig, they appear to be square-shaped. Wouldn't it make more sense for the patches to be a function of the topography of even the drainage network itself?*

We now say "These are rectangular in shape with K values that taper to the baseline $K$ over ten pixels. We acknowledge this pattern is not very realistic but the aim is not to recreate real landscapes but rather to confuse the algorithms for quantifying concavity and test if they can still detect modelled concavity even if we violate some of the assumptions implicit in the concavity algorithms."

*Page 12, line 3: 'reference concavities between 0.4 and 0.5 should give an accurate representation of the relative steepness'. Do you mean that in general or just for the Loess Plateau? If generally, then it calls for a justification. How does it relate to your natural basalt-sandstone experiment in Oregon?*

Added phrase "in this area of the Loess Plateau" to make it clear we are only referring to this study site.

*Page 12, lines 3-10: repeated text.*

Fixed.

*Page 13, line 2: A short discussion of how the lithology is expected to affect m/n is probably needed here. (Possibly via the relation between channel width and specific stream power/drainage area?)*

We don't actually know this but we can put some speculation here. For example the different incision rules for plucking and fluting have different predicted slope exponents so we can attempt to connect that to concavity. As far as we are aware, however, nobody has attempted a systematic investigation of how lithology affects concavity. In fact, this is something we would like to do once we (and reviewers) are confident in our methods.

*Page 13, lines 15-16: Could be worth mentioning that the Gulf of Evia overall represents a natural experiment where U varies both temporally and spatially.*

We liked this description so have used almost exactly this phrase in the paper. Thanks.

*Page 14, lines 15-18: How exactly does drainage area change affect the derived m/n? If all the tributaries are losing area, then they should all be plotted as convex in the chi-z domain. But the technique tries to minimize the residual and not to straighten the profiles. How is the residual affected by area change?*

The methods do not account for drainage area change so we will state this in the text to avoid confusion and also mention how drainage area change may lead to convexities or concavities in the chi plots that will cloud interpretation of concavity.

*Page 14, line 21: 'bust'*

Fixed.

*Page 15, lines 13-16: This appears to be a key sentence, but its relation to the results and discussion is not straightforward.*

We will try to be more clear in the revision. What we are trying to say is that if you get concavities from many small basins you may be able to build up some sense of regional variations in concavity, and these might be linked (empirically) to things like climate, tectonics, and lithology. We are attempting to foreshadow what we intend to do with these algorithms in the future, and what we hope other workers might use our algorithms to do.

*Fig 7: maybe it's worthwhile explaining what are the squared low relief patches in the variable K panels.*

We will add this to the caption.

*Fig 9: The captions of panel C are not clear. The two chi-based methods have different m/n maxs.*

We will clarify this.

*Fig 11: I assume that the dashed line represents faults. Maybe add a legend. Also, it might be worth differentiating (by color) between basins that drain across relay ramps and those that drain across faults.*

Good advice. We will do this.

*Fig 12: Same comment: differentiate between basins that drain across relay ramps and those that drain across faults.*

Good advice. We will do this.

*Fig 13: From my experience in chi-z analysis, such a scatter and concave tributaries are indicative that the chosen m/n is too high. Can you show the same basin with different m/n. This might hint that the scatter minimization technique and your new MLE technique give different results.*

Our plotting functions as a matter of course print out the chi-elevation profiles of every basin for every $m/n$ ratio (and also produce mpeg files showing how the chi-z plots change as $m/n$ ratio increases) so it is fairly easy for users to check if the algorithm is producing results that raise alarm bells. Figure 3 shows other $m/n$ ratios for this basin.

*Wang 2017b probably deserves more credit for comparing the chi-z to slope-area predictions.*

Yes, we agree. We will add more text highlighting this contribution.

[Figure]

**Fig. 1.** Best fit m/n ratios for variable uplift scenario (from Figure 7 in the discussion paper).

[Figure]

Best fit *m/n*

1.0

0.8

0.6

0.4

| | Chi Monte Carlo |
| | Chi all data |
| | Chi disorder |
| | S-A all data |
| | Segmented S-A |

0    2    4    6

Basin key

**Fig. 2.** Best fit m/n ratios for variable erodibility scenario (from Figure 7 in the discussion paper).

**Fig. 3.** Chi-elevation plots of Evia basin 7 (from Figure 13 in the discussion paper).

---

## Author Comment (AC3) · 21 Apr 2018

We thank the associate Editor (Jens Turowski) for steering this paper through such a helpful and efficient review process (with two detailed and useful reviews).

*We have now received two generally positive reviews of the paper. The issues raised by the reviewers seem self-explanatory and fairly straight-forward to deal with, and I do not think that I need to elaborate on them. I would just like to highlight a small comment by reviewer 1: there currently is a slight mismatch between the title and the content of the paper. By just reading the title, the reader may not expect a methodological paper, and the question currently posed in the title is not actually addressed in the paper. I encourage you to re-think the title such that it better reflects the content and aims of the paper.*

[Figure]

We see the rationale behind this comment and are sad because the current title is more fun than "Extraction of concavity and collinearity from channel networks", which is probably what we will use in the revision.
* * *
**ESurfD**

---

## Author Response (AR1)

THE UNIVERSITY of EDINBURGH
School of Geosciences

Simon M. Mudd
*School of Geosciences*
*University of Edinburgh*
*Drummond Street*
*Edinburgh, EH8 9XP*
*Phone: +44 (0)131 650 2435*
*Email: simon.m.mudd@ed.ac.uk*

Jens Turowski
Associate Editor, Earth Surface Dynamics

May 23, 2018

Dear Dr. Turowski,

Thank you for considering our manuscript 'How concave are river channels?'. We are grateful to the reviewers for providing constructive feedback and allowing us to improve the manuscript.

We have made significant changes to our manuscript following the comments we received.

Please find below detailed responses to the individual points raised by each of the reviewers, along with a version of our manuscript highlighting the changes we have made to answer the reviewer comments. Throughout our responses we refer to line numbers in our manuscript: these are the correct line numbers in the manuscript with the changes incorporated. We have endeavoured to address all concerns and return the manuscript in a publication-ready state.

One thing to note is that since our online response, we have done some more digging into the literature and have come around to the opinion that one *can* use collinearity as a basis to judge concavity. That is, the collinearity tests can be performed independent of stream power. We have therefore retained the title from the original manuscript but now make very little reference to $m/n$ ratios and instead focus on concavity. Stream power is still introduced, since it underpins our numerical models, but all topographic data from real landscapes now refers to channel concavity.

In addition to the major change of focusing on concavity rather than SPIM exponents, we have also i) implemented another $\chi$–based method of estimating the most likely concavity ii) updated all figures to include this method and to replace $m/n$ with $\theta$ iii) Updated figures at the Evia site to better show the faults and basins affected by fault relays.

We feel these changes have significantly improved the paper and again thank the reviewers for their suggestions.

In the responses below, *the reviewer comments are in italics* and our responses are in plain text.

Sincerely,

Simon M. Mudd

**AE comments**

*We have now received two generally positive reviews of the paper. The issues raised by the reviewers seem self-explanatory and fairly straight-forward to deal with, and I do not think that I need to elaborate on them. I would just like to highlight a small comment by reviewer 1: there currently is a slight mismatch between the title and the content of the paper. By just reading the title, the reader may not expect a methodological paper, and the question currently posed in the title is not actually addressed in the paper. I encourage you to re-think the title such that it better reflects the content and aims of the paper.*

In our response we uploaded a month ago we were still connecting stream power to the estimates of concavity but after some more derivations and reading we have decided that we can link collinearity of tributaries to geometric concavity which was the recommendation of reviewer 1. In this case, our entire discussion relates to concavity rather than parameters of stream power and we have replaced mention of the $m/n$ ratio in much f the paper with references to concavity. We still mention stream power since this is what drives the numerical simulations. However the paper is now focused on the concavity rather than exponents of the SPIM. We therefore have elected to not change the title since we feel it now does reflect the contents of the paper.

**Reviewer 1**

We thank reviewer 1 (Roman DiBiase) for his thorough review and highlighting a different way of casting the paper that does not rely on stream power. We will still make some mention of stream power because it serves as the basis for numerical simulations, and also plays a role in the assumption of collinearity (see below), but we take the advice that introducing the concept of concavity can be done without this restrictive assumption. These reviewer comments have very much helped make the context of the paper more general, and thank the reviewer for these suggestions which we feel have substantially improved the paper.

*This paper presents a new method for constraining the intrinsic concavity of river channels, in order to more accurately interpret spatiotemporal patterns of climate and tectonics from landscapes that deviate from the simpler case of steady state, uniform rock uplift, rock strength, and climate. The new metric compares the chi-elevation plots of tributary and mainstem channels in an objective manner, and is integrated into LSDTopoTools, an open source topographic analysis environment developed by the authors. This paper then evaluates the model as compared to existing approaches, using examples from real and synthetic landscapes. Overall, this is a nicely-written paper with great figures and the code seems like a very useful addition to an arsenal of topographic analysis scripts that have evolved in recent years (e.g., LSDTopoTools and TopoToolbox). I think this paper fits well at ESurf, and I only have one major issue that I think needs to be resolved before publication:*

Thank you for your supportive comments. As we describe below, we agree with the suggested revision (see below) and will carry it out in the revision.

*Major comment: On Page 4, Line 25, the authors recognize a strength of the existing slope-area method of determining channel concavity is that it requires no assumptions whatsoever about the underlying form of the equations describing channel incision. Thus, I was surprised to find that the chi analysis underpinning the new method was (unnecessarily) framed in terms of the stream power model! Although the Perron and Royden 2012 paper also frames chi in terms of stream power, I would instead recast equations 7-9 in terms of the more general empirical relationship of Flints law (equation 1), which makes no assumptions about process ks and theta are simply geometrical properties of river channels. We did this in Whipple et al. 2017 Geology (doi:10.1130/G38490.1), but did not expand too much on*

*the reasoning.*

In our first response we agreed with this comment but we thought that our metric for the correct concavity using chi analysis was collinearity and at that time through it would be difficult to justify separating this from the $m/n$ ratio. We have changed our minds about this after careful study of both the Niemann et al. (2001) paper and the Wobus et al., (2006) paper. We now feel that collinearity can be connected to concavity (in the sense of Flint's law) and we have completely rewritten the introduction to reflect this. There is a new section **Connecting concavity to collinearity** where we argue that collinearity tests used in chi analysis can be related to the concavity values that one might extract from slope area data. The entire context of the paper has now changed to move away from stream power and toward purely geometric considerations.

*Note also that the relationship between channel steepness and erosion rate/uplift rate (Page 3, Line 21-29) is again not necessarily tied to stream power, but relates to an empirical relationship between relief and erosion rate (equation 1 of DiBiase and Whipple, 2011, doi:10.1029/2011JF002095; also discussed in Whipple and Meade 2006, doi:10.1016/j.epsl.2005.12.022). Connecting this exponent and the concavity index to ms and ns in stream power gets problematic because things vary depending on the specific form of the incision law (for example, adding a threshold changes the steepness-E relationship without changing m or n).*

We now specifically highlight these in the revised introduction:
"A number of studies (e.g., Ouimet et al., 2009; DiBiase et al., 2010, Scherler et al., 2014, Harel et al., 2016) have demonstrated that $k_s$ is positively correlated with erosion rate, mirroring the predictions of Gilbert (1877) over a century earlier."

*I think the paper would be stronger if, like the title says, the main analysis focuses on finding the intrinsic concavity index theta, rather than the model-dependent ratio m/n. Note that this of course does not preclude the comparison with stream power model landscapes shown in section 3 and interpretation/comparison with expected m/n!*

See above. We have completely rewritten the introductory materials to reflect this comment.

*Page 5, Line 16: I think only the profile is smoothed, rather than the full DEM.*

The Wobus paper actually recommends smoothing the DEM: it was written in the dark ages of DEM quality. However we now note that modern workers don't do this.

*Page 5, Line 23: Is method (i) using a single channel, the entire channel network? Whole DEM?*

Clarified in the text: it uses all the tributaries and the main stem in a given basin.

*Page 7, Line 1: This is just one new method, correct?*

We now call it two (there is the all points and what we were calling the "monte carlo points" methods. Liran Goren suggested we call the second a bootstrap method.

*Page 7, Line 9-10: Not totally necessary, but might be helpful to emphasize the MLE = 1 for r = 0.*

Done.

*Page 7, Line 11: There seems to be a mistake in the math here where it was assumed that exp(ab) = exp(a)exp(b) rather than exp(a)b.*

Thanks for spotting that. We inserted this mistake as a rhetorical device and it doesn't affect the results. We have expunged this equation from the manuscript.

*Page 8, Line 5-9: Not just hanging tributaries, but any complexities influencing concavity that are not captured by simple stream power framework (e.g., spatial patterns in sediment cover/grain size). Perhaps it makes sense to include areas upstream of these hanging tributaries in the statistical analysis? Maybe collinearity is too stringent, and similar steepness is instead more useful?*

A local linearity test requires some segmentation process (which is what some of the authors of this paper tried in Mudd et al. 2014 and we find that method is extremely noisy and uncertain. We have tried to highlight the drawbacks of collinearity but we feel its advantages outweigh its disadvantages (we now say this in the conclusion, and explain why we feel this way).

*Page 9, Line 34: i) by regression of all chi-elevation data Make clear whether this is just one channel or the whole tributary network at once*

We now say "For all but the final method the analyses use all tributaries in the basins."

*Page 12, Line 3-10: Typo: This text is directly repeated from above.*

Fixed.

*Page 12, Line 32: Note that Duvall et al. (2004) argue that the high concavities in the Santa Ynez Mtns are due to strong rocks in the headwaters and weak rocks below, which is different than the "spatially varying m/n as a function of lithology" shown in Fig. 10.*

We now say: "Duvall et al. (2004) suggested that having hard rocks in headwaters and weak below might influence concavity and this and other hypothesis could be tested by comparing concavities in both monolithologic basins and basins with mixed lithology."

*Page 13, Line 19-20: I agree - but then why is it appropriate to use this for the numerical experiment on landscape transience, which also includes knickpoints?*

In practice, workers generally fit small sections of the channel network with a concavity because the knickpoints distort the overall concavity. This is typically done in a totally ad-hoc manner. The tutorials and code associated with the Wobus et al. (2006) paper, for example, include functions to let users manually choose intervals over which to select concavity. One of our main goals is to ensure reproducibility, so we attempted to use a segment finding algorithm (mentioned in the paper). This sometimes works, and sometimes doesn't. So we find it very difficult to select appropriate segments for concavity using S-A data with reproducible techniques–this is true for both numerical simulations as well as in real landscapes. I suppose we are setting up a straw man for the numerical simulations, but this straw man situation is the one every geomorphologist finds themselves in when they are doing S-A analysis.

*Page 13, Line 23: I think more importantly, other processes become important in the transient! (e.g., DiBiase et al, 2015, doi:10.1130/B31113.1)*

We now say: "A number of authors have suggested that in both highly transient and rapidly eroding landscapes processes other than fluvial incision become important in shaping the channel profile, such as debris flows and plunge pool erosion (Stock and Dietrich, 2003, DiBiase et al., 2015, Scheingross and Lamb, 2017)."

*Page 14, Line 7: Spatial gradients in tectonics are far more important than temporal variations in disrupting interpretations of chi at divides. if spatially uniform U/K, then chi still good indicator of divide instability during temporally varying U (or K).*

We now add the sentence: "On the other hand, numerical simulations suggest that spatial variability in uplift are more important that temporal gradients in uplift rates (Whipple et al., 2017)."

*Page 14, Line 9-18: I dont quite agree here. The fact that this is a relay system means that spatially variably uplift likely dominates, complicating a simple interpretation of chi across divides (see Whipple et al., 2017 JGR, doi:10.1002/2016JF003973)*

We have simply deleted this sentence since this paper is not about tectonics of Evia: we have used it merely to highlight that one can use the concavity code in tectonically complex areas. By deleting this sentence we stick to uncontroversial observations of the topography and the chi coordinate.

*Page 14, Line 19-22: ...river profiles...are not alone sufficient to interpret the history of landscape evolution, but must be considered alongside other observational data and in the context of a process-based understanding of landscape evolution ... I strongly agree!*

We are optimistic that a richer set of metrics will be used in the future as topographic and other data improves.

*Page 14, Line 21: Typo bust*

Fixed.

*Page 14, Line 32: Be careful tying the paper to stream power! (see main comment above)*

We have removed almost all method of SPIM except for components when we are referring to model results that are driven by the SPIM.

*Page 15, Line 4-6: I think would be good to point out that the second method does not handle well spatially variable rock uplift rate.*

We don't say this, but instead say the disorder metric is the most tightly constrained.

*Figure 1: More detail is needed in caption to explain this sketch. Is it a single trunk channel? An entire stream network? There is also some good discussion of these challenges of interpreting concavity in Gasparini and Whipple, 2014 (doi:10.1130/L322.1).*

We have updated the caption, specifically referring to the interpretations labeled in the plot, so

that it is more clear. We also added a sentence and cited the Gasparini and Whipple paper: "Conversely, if a single reference concavity is chosen in an area with changing concavity, then spurious patterns in in $k_{sn}$ may arise (e.g., Gasparini and Whipple, 2014)."

*Figure 2: Again, is this a single channel? Whole tributary network?*

Clarified. We now say "The data is taken from only the trunk channel."

*Figure 3: This caption could use more description. Hard to follow without careful reading of main text.*

We have rewritten the caption. Hopefully it is clearer now.

*Figure 11: Do you mean UTM Zone 34N?*

Fixed.

**Reviewer 2**

We thank reviewer 2 (Liran Goren) for a number of helpful comments that will help improve the paper. Reviewer 2 is entirely correct that we should have tested the disorder metric that has been used in several recent papers. We have followed this advice and we find that it performs similarly to the "all chi" method and in some cases is the most accurate method. It is also more computationally efficient than other methods. The only real drawback is that because it uses all data it cannot express the uncertainty in the concavity, so we now recommend using either the disorder and Monte Carlo point method, or all three $\chi$ methods to extract concavity.

*The manuscript presents and compares several techniques for extracting the concavity index of fluvial basins from topographic fluvial data. The manuscript nicely states how, for different (yet, specific) models of fluvial incision, the true, process-dependent (or process-assumed), concavity index is a crucial parameter, without which, the steepness index and information about time and space dependent uplift rates cannot be reliably retrieved. The importance of the concavity index and the motivation behind the presented analyses are therefore convincing.*

*The manuscript is well written, and the effort that was invested in articulating the scope of the problem and the different techniques and analyses eases the reading of even complicated concepts.*

Thanks. We are glad to hear that the manuscript is clear.

*Overall, the manuscript compares between two classes of techniques for extracting the concavity index, slope-area analysis and chi-z analysis. Through several insightful numerical examples the superiority of the chi-z analysis is demonstrated in particular for spatially heterogeneous and transient landscapes. The manuscript then turns to explore the concavity index of natural landscapes, where the conclusions are, as expected, more ambiguous.*

Natural landscapes are indeed a vexing problem since there is no way to know the "real" concavity, although as reviewer 1 notes we can slightly reduce this confusion by clarifying that the method aims to constrain the geometric concavity rather than parameters that assume some form of the physics of incision. Interpreting these data will continue to confound workers, as the reviewer here clearly points out!

*I have one major concern: Given that the manuscript is methodological in nature, namely, it explores the accuracy and robustness of different techniques for evaluating the concavity index, it is lacking essential reasoning for developing a new technique without exploring existing ones or even just pointing out their possible theoretical limitations. Here, I specifically refer to the development of the maximum likelihood estimator for m/n from chi analysis (which is split into two techniques), without exploring existing techniques such as the tributary scatter reduction (Goren et al., 2014) and a later version of this technique developed in Hergarten et al., 2016 (both papers are cited in the manuscript). These techniques find the m/n that minimizes the scatter in elevation over chi bins. They are intuitive, computationally simple, and the scatter itself can be used to evaluate the uncertainty. Developing a new technique that appears to be computationally more demanding without comparing and contrasting it to existing techniques does not serve the goals of the manuscript and of the community that can benefit from it.*

We agree, this was an oversight. We have implemented the disorder metric and tested it against all our landscapes. It does quite well! Throughout the manuscript you will see the results from this method now.

*On the same note, I would like to draw the authors attention to a pre-print https://eartharxiv.org/5u9eg/ (recently accepted for publication in JGR-ES) that, for a different geomorphic application, compares m/n values derived from slope-area and from chi-z using the tributary scatter reduction technique. I'm a co-author on this manuscript and I apologize for this far from elegant self-promotion, but its very relevant to the current manuscript under discussion.*

We are delighted in this self promotion since the paper is very interesting and we are happy to have it brought to our attention. The findings in that study are relevant to our work and we now cite it in the paper in the section on calculating the disorder metric as well as in the discussion about the Evia catchments.

*Another, more minor, comment, is that currently, the manuscript is missing a discussion about which and under what conditions each of the two chi-based techniques for extracting m/n is better.*

We now say: "We find that $\chi$-based methods are best able to reproduce the concavity values imposed on the model runs. We recommend users calculate the most likely concavities using the bootstrap and disorder methods as these provide estimates of uncertainty, although the disorder method is the most tightly constrained of the $\chi$-based methods."

*Page 3, line 4: Within the scope of the current manuscript the adjective constant for m and n is a bit misleading.*

Deleted the word "constant".

*Page 6, line 9: The chi coordinate is simply a derived function of topography. Its a function of the distribution of the drainage area, or the topology, and not of the topography.*

This sentence no longer appears since we have reorganised how we introduce chi by integrating Flint's law, as requested by the first reviewer. But thanks for pointing this out because we probably would have said it in another paper so thanks for correcting us.

*Page 7, lines 15-17: The technique of minimizing z scatter over chi bins that was mentioned above does*

*not have this issue.*

The disorder statistic will still have this issue because longer tributaries will diverge from the trunk channel more so will add more weight to the disorder statistic. However we have implemented a technique to estimate uncertainty by using all combinations of tributaries.

*Page 7, lines 22: Could it be that bootstrapping is a more accurate description than Monte-Carlo?*

Yes, you are correct. We have changed the name.

*Page 8, line 13: must*

Fixed.

*Page 10, line 19: The geometry of the K patches should be described. From the fig, they appear to be square-shaped. Wouldnt it make more sense for the patches to be a function of the topography of even the drainage network itself?*

We now say "These are rectangular in shape with $K$ values that taper to the baseline $K$ over ten pixels. We acknowledge this pattern is not very realistic but the aim is not to recreate real landscapes but rather to confuse the algorithms for quantifying concavity and test if they can still detect modelled concavity even if we violate some of the assumptions implicit in the concavity algorithms."

*Page 12, line 3: reference concavities between 0.4 and 0.5 should give an accurate representation of the relative steepness. Do you mean that in general or just for the Loess Plateau? If generally, then it calls for a justification. How does it relate to your natural basalt-sandstone experiment in Oregon?*

Added phrase "in this area of the Loess Plateau" to make it clear we are only referring to this study site.

*Page 12, lines 3-10: repeated text.*

Fixed.

*Page 13, line 2: A short discussion of how the lithology is expected to affect m/n is probably needed here. (Possibly via the relation between channel width and specific stream power/drainage area?)*

We added some text here: "Whipple and Tucker (1999) suggested that concavity is controlled primarily by discharge–drainage area and channel width–drainage area relationships, which may be influenced by lithology, but other authors have found systematic variations in concavity with lithology (e.g., Duvall et al., 2004, van Laningham et al., 2006, Lima and Flores 2017). Lima and Flores (2017) suggested that the thickness of basalt flows could influence concavity, with different knickpoint propagation mechanisms in massive versus thinly bedded flows."

*Page 13, lines 15-16: Could be worth mentioning that the Gulf of Evia overall represents a natural experiment where U varies both temporally and spatially.*

We liked this description so have used almost exactly this phrase in the paper. Thanks.

*Page 14, lines 15-18: How exactly does drainage area change affect the derived m/n? If all the tributaries are losing area, then they should all be plotted as convex in the chi-z domain. But the technique tries to minimize the residual and not to straighten the profiles. How is the residual affected by area change?*

Based on comments of the other reviewer we have deleted these sentences.

*Page 14, line 21: bust*

Fixed.

*Page 15, lines 13-16: This appears to be a key sentence, but its relation to the results and discussion is not straightforward.*

We now say: "Thus we hope future workers can calculate reliable, reproducible concavity values for many small basins in regions with spatially varying uplift, climate or lithology to test if patterns in concavity can be linked to variations in these landscape properties."

*Fig 7: maybe its worthwhile explaining what are the squared low relief patches in the variable K panels.*

We now say: "The rectangular patches of low relief are area of high erodibility in the left column."

*Fig 9: The captions of panel C are not clear. The two chi-based methods have different m/n maxs.*

We have updated the caption to make it clear that the probability distribution is of all most likely concavity values rather than uncertainty in an individual basin.

*Fig 11: I assume that the dashed line represents faults. Maybe add a legend. Also, it might be worth differentiating (by color) between basins that drain across relay ramps and those that drain across faults.*

Done. It looks much better. Thanks.

*Fig 12: Same comment: differentiate between basins that drain across relay ramps and those that drain across faults.*

Done.

*Fig 13: From my experience in chi-z analysis, such a scatter and concave tributaries are indicative that the chosen m/n is too high. Can you show the same basin with different m/n. This might hint that the scatter minimization technique and your new MLE technique give different results.*

We have included chi profiles across the different concavity values.

*Wang 2017b probably deserves more credit for comparing the chi-z to slope-area predictions.*

Yes, thank you for highlighting this. In the introduction we now say:

[revised manuscript text omitted]